# Higher-group symmetry in finite gauge theory and stabilizer codes

**Maissam Barkeshli[1], Yu-An Chen[1,2], Po-Shen Hsin[3], Ryohei Kobayashi[1]**

[1] Department of Physics, Condensed Matter Theory Center, and Joint Quantum Institute, University of Maryland, College Park, Maryland 20742, USA

[2] Joint Center for Quantum Information and Computer Science, University of Maryland, College Park, Maryland 20742, USA

[3] Mani L. Bhaumik Institute for Theoretical Physics, 475 Portola Plaza, Los Angeles, CA 90095, USA

### Abstract

A large class of gapped phases of matter can be described by topological finite group gauge theories. In this paper we show how such gauge theories possess a higher-group global symmetry, which we study in detail. We derive the $d$-group global symmetry and its 't Hooft anomaly for topological finite group gauge theories in $(d+1)$ space-time dimensions, including non-Abelian gauge groups and Dijkgraaf-Witten twists. We focus on the 1-form symmetry generated by invertible (Abelian) magnetic defects and the higher-form symmetries generated by invertible topological defects decorated with lower dimensional gauged symmetry-protected topological (SPT) phases. We show that due to a generalization of the Witten effect and charge-flux attachment, the 1-form symmetry generated by the magnetic defects mixes with other symmetries into a higher group. We describe such higher-group symmetry in various lattice model examples. We discuss several applications, including the classification of fermionic SPT phases in (3+1)D for general fermionic symmetry groups, where we also derive a simpler formula for the $[O_5] \in H^5(BG, U(1))$ obstruction that has appeared in prior work. We also show how the $d$-group symmetry is related to fault-tolerant non-Pauli logical gates and a refined Clifford hierarchy in stabilizer codes. We discover new logical gates in stabilizer codes using the $d$-group symmetry, such as a Controlled-Z gate in (3+1)D $\mathbb{Z}_2$ toric code.

# 1   Introduction

Topologically ordered phases of matter can be characterized in general by intricate patterns of fusion and braiding of topologically non-trivial defects (excitations). In two spatial dimensions, topologically non-trivial quasi-particles can be thought of as point-like, codimension-2 topological defects, and their universal properties are characterized by a unitary modular tensor category [1, 2, 3, 4, 5, 6, 7, 8, 9]. Over the past decade, it was understood that codimension-1 topological defects in (2+1)D also can possess non-trivial topological properties [10, 11, 12, 13, 14, 15, 16, 17, 18, 19]; a full account of the universal properties of (2+1)D topological phases of matter must therefore include both codimension-1 and codimension-2 defects. The fusion and braiding properties of codimension-1 and codimension-2 defects together form a more complicated mathematical structure – a unitary fusion 2-category [20, 11, 21]. This fusion 2-category plays a critical role in the modern understanding of symmetry-enriched topological orders in terms of $G$-crossed braided tensor categories [22, 23, 24, 25, 26].

In general, $(d+1)$ space-time dimensions, much less is known about the general algebraic structure of topological defects, aside from a general expectation that they should be characterized by a unitary fusion $d$-category. A complete understanding of topological defects is a difficult question, in part because the problem of classifying topological defects contains within it the problem of classifying all topological phases of matter. The more tractable problem is to focus on the structure of invertible topological defects, as the classification of invertible topological phases is a discrete Abelian group in each dimension [27, 28, 29, 30, 31, 32, 33, 34, 35, 26].

Over the past several years, the relationship between topological defects of varying codimension and symmetry in quantum field theory has come into increasingly sharp focus. Invertible topological defects of codimension-$k$ define a $(k-1)$-form symmetry of the field theory [36, 37, 38]. The interplay between invertible topological defects of varying codimension form the structure of a $d$-group symmetry [22, 39, 23, 40, 41] in $(d+1)$ space-time dimensions. Higher-group symmetries are present in many quantum systems, and they have applications to topological phases, dynamics and vacuum structure in quantum chromodynamics, spin liquids, hydrodynamics, holography, higher-dimensional critical systems, Higgs and axion physics, and conjectures in quantum gravity, see *e.g.* Refs. [39, 23, 42, 43, 40, 41, 44, 45, 46, 47, 48, 49, 50, 51, 52, 53, 54, 55, 56, 57]. The non-invertible topological defects (simple examples being non-Abelian anyons in (2+1)D) define higher categorical (non-invertible) symmetries, as studied in *e.g.* Refs. [58, 59, 60, 61, 62, 63, 64, 65, 66, 67, 68, 69], which are even more exotic. Therefore understanding the higher group structure of invertible topological defects amounts to understanding the higher-group symmetry of the field theory.

One large class of gapped phases of matter can be described by topological finite gauge theory, that is, gauge theory with a finite, discrete gauge group. Such theories can be studied using exactly solvable lattice models [70, 71, 72, 73, 74, 75]. In (3+1)D, it has been argued that all topologically ordered phases of matter with fully mobile excitations can be described by finite gauge theory coupled to either bosonic or fermionic point charges [76, 77, 78]. While the electric and magnetic defects in finite gauge theories are well known and have been studied in some depth [79, 80, 81, 82], the full higher-group structure of invertible topological defects in finite gauge theories has not been developed.

Recently some progress was made in understanding part of the 3-group symmetry in (3+1)D gauge theories with finite gauge group [44, 83]. In this paper, we continue this line of work and develop a more comprehensive understanding of the $d$-group symmetry in $(d+1)$D finite gauge theory with gauge group $G$, together with its 't Hooft anomaly. In addition to including both Abelian and non-Abelian $G$, we include the possibility of a Dijkgraaf-Witten twist for the gauge theory [82], characterized by a cohomology class $[\omega^{(D)}] \in H^D(BG, U(1))$.

The $d$-group symmetry arises essentially from the following phenomena. In theories with bosonic charges, the higher form symmetries are generated by decorating submanifolds with symmetry-protected topological (SPT) phases and then gauging the $G$ symmetry of the full theory. The magnetic codimension-2 defects, when intersecting the gauged SPT defects, will source lower dimensional gauged SPT defects. An example of this was described in detail recently in Refs. [44, 83]: in (3+1)D, the intersection point between magnetic strings and (1+1)D gauged SPT defects sources an electric charge. The generalization to magnetic defects crossing generic gauged SPT defects and sourcing lower-dimensional gauged SPT defects can be viewed as a generalization of this charge-flux attachment and the Witten effect [84] on submanifolds. See Ref. [85] for a generalization to defects in nonlinear sigma models.

Moreover, in the presence of a Dijkgraaf-Witten twist $\omega^{(D)}$, there are several non-trivial changes to the properties of the magnetic defects, which we discuss in this paper. First, the magnetic defects become dressed with gauged SPT defects of one higher dimension, which is another analog of charge-flux attachment. This dressing implies

that some of the magnetic defects can become non-Abelian, even when associated with Abelian magnetic flux, and become endowed with exotic non-Abelian braiding transformations. This implies that the invertible magnetic defects correspond to a subgroup $Z_\omega(G) \subset Z(G)$ of the center $Z(G)$. Second, the fusion rules of the magnetic defects themselves get modified such that they can fuse to gauged SPT defects.

On the lattice, the higher-group symmetry manifests as commutation relations that do not produce a number, but a non-trivial operator: if the generators of symmetries are the operators $\mathcal{O}_1, \mathcal{O}_2, \mathcal{O}_3$ acting on the Hilbert space, an example of a higher group structure is

$$[\mathcal{O}_1, \mathcal{O}_2] \equiv \mathcal{O}_1 \mathcal{O}_2 \mathcal{O}_1^{-1} \mathcal{O}_2^{-1} = \mathcal{O}_3 . \tag{1}$$

This implies that $\mathcal{O}_3$ is created from the configuration of generators on the left. We can also generalize the relation to a nest of commutators of the form $[\mathcal{O}_1, [\mathcal{O}_2, [\mathcal{O}_3, \cdots]]] = \mathcal{O}'$ . This is to be contrasted with the case where the commutation relation produces a number (different from unity), which is the same in any state, and represents an 't Hooft anomaly of the symmetry that remains the same across different states (and thus same for states of different energies); here, the expectation value of the commutation relation is different on states with different eigenvalues of $\mathcal{O}_3$. Such "operator-valued anomaly" means the symmetry algebra is modified to be a higher group, and is by definition not an 't Hooft anomaly, as emphasized in Refs. [40, 41]. Examples of lattice gauge theory models with higher-group symmetry are discussed in Ref. [83].

In general, the symmetry generators of finite Abelian gauge theory are realized as the logical gates on the codespace in a stabilizer code, which is the ground state subspace of some stabilizer lattice Hamiltonian [86, 87, 88, 89, 90, 91, 92]. When the symmetry generators correspond to the logical gates, the higher-group symmetry that mixes the symmetry generators endows a non-trivial relation between the logical gates. For instance, we will encounter examples where $[\mathcal{O}_2, \mathcal{O}_3] = 1$, and together with $[\mathcal{O}_1, \mathcal{O}_2] = \mathcal{O}_3$ they imply that at least one of $\mathcal{O}_1, \mathcal{O}_2, \mathcal{O}_3$ must realize a logical gate that cannot be described by the Pauli operator. In other words, the higher-group symmetry of the stabilizer Hamiltonian model implies that the corresponding stabilizer code has a non-Pauli logical gate. As concrete examples, we find that the generator of 0-form symmetry in (3+1)D $\mathbb{Z}_2$ and $\mathbb{Z}_2^2$ toric code, in general, realizes the non-Pauli Clifford gate, which is enabled by the non-trivial 3-group structure among 0-form, 1-form and 2-form symmetries. In particular, we construct control-$Z$ gate in (3+1)D $\mathbb{Z}_2$ toric code. We also derive a new upper bound on the possible logical gates implemented by the gauged SPT defects of untwisted $\mathbb{Z}_2^N$ gauge theory in generic dimensions, which refines the known result given in Ref. [93].

An important property of global symmetry is its 't Hooft anomaly, *i.e.* obstruction to gauging the symmetry. The 't Hooft anomaly constrains the boundary properties and phase transitions between different gapped phases, and it is also useful in constraining the low energy dynamics of the microscopic model, such as ruling out symmetric gapped phases in quantum systems with an anomaly that cannot be matched by the proposed low energy dynamics, see *e.g.* Refs. [44, 94, 57]. In particular, it has been understood that 't Hooft anomalies are intimately related to Lieb-Schulz-Mattis constraints [95, 96, 97] in many-body quantum systems [98, 99, 100, 101]. There is also a conjecture that two quantum systems with the same anomaly for all symmetries should be dual to each other, or can be connected by a symmetry preserving continuous deformation.

One important application of understanding the $d$-group symmetry and its 't Hooft anomaly is in classifying SPT states and also symmetry-enriched topological orders. For example, consider the problem of classifying fermion SPT states with fermionic symmetry group $G_f$, which is a central extension of a bosonic symmetry group $G_b$ by fermion parity $\mathbb{Z}_2^f$. Gauging the $\mathbb{Z}_2^f$ fermion parity gives a "bosonic shadow" theory, which involves a dynamical $\mathbb{Z}_2^f$ gauge field coupled to fermionic matter, with a global $G_b$ symmetry [102, 103, 104, 105, 106]. Understanding how to characterize and classify such $\mathbb{Z}_2^f$ gauge theories with $G_b$ symmetry is intimately related to the problem of characterizing and classifying fermion SPT states with $G_f$ symmetry. Furthermore, the former requires an understanding of the $d$-group symmetry and its 't Hooft anomaly in $\mathbb{Z}_2^f$ gauge theory. In this paper, we will follow this program and develop a classification of (3+1)D fermion SPTs with general $G_f$ symmetry. Our results give an alternative perspective on the classification derived previously in Refs. [33, 34]. Using our methods, we also are able to give a significantly simpler formula for the $[O_5] \in H^5(BG, U(1))$ obstruction appearing in the classification of (3+1)D fermionic SPTs.

This paper is organized as follows. In Section 2, we discuss the generalized Witten effect, that dresses magnetic defect with the generalization of the "electric charge" in the presence of topological interaction. In Section 3, we investigate the higher-group symmetry in examples such as Abelian gauge theory, using the generalized Witten effect and also exactly solvable lattice Hamiltonian models. In Section 4, we investigate the anomaly of the higher-group symmetry in these examples. In Section 5, we discuss the application of higher-group symmetry in lattice models and the relationship to fault-tolerant logical gates that are not Pauli operators. In Section 6 and 7, we investigate higher-group symmetry and its 't Hooft anomaly in general finite group gauge theories with bosonic electric charges. In Section 8, we investigate the 3-group symmetry in $\mathbb{Z}_2$ gauge theory with fermion particles in (3+1)D, and discuss a

new construction of fermionic symmetry-protected topological (SPT) phases in (3+1)D and their classification using the 3-group symmetry.

There are several appendices. In Appendix A we summarize some mathematical properties of an $n$-group. In Appendix B we discuss higher-group symmetry in Abelian gauge theories by embedding the gauge fields into continuous $U(1)$ gauge fields. In Appendix C we give some details of the data classifying the fermionic SPT phases in (3+1)D.

# 2 Generalized Witten effect and charge-flux attachment

In this section, we consider $G$ gauge theory in $D$ space-time dimensions with topological action $S[a] = \int_{M_D} a^* \omega^{(D)}$ for the $G$ gauge field, and we will take $G$ to be a finite group, which can be Abelian or non-Abelian. The partition function on space-time manifold $M_D$ takes the form

$$Z(M_D) = \frac{1}{|G|} \sum_{[a]} e^{i \int_{M_D} a^* \omega^{(D)}}. \tag{2}$$

Here $[\omega^{(D)}] \in H^D(BG, U(1))$, $\omega^{(D)}$ is a representative $D$-cocycle, $a$ is the dynamical 1-form $G$ gauge field, $a^* \omega^{(D)}$ is the pullback to a representative cocycle for $H^D(M_D, U(1))$, and the sum is over all flat principal $G$ bundles over spacetime $M_D$. On $D$-dimensional sphere, the bundles are trivial, and the partition function is $1/|G|$.

The theory has codimension-2 magnetic defects, around which there is a non-trivial holonomy of the $G$ gauge field that takes value in conjugacy class $[g]$, see *e.g.* Ref. [107]. For finite group $G$, these magnetic defects are topological.

One can also consider gauge theories with continuous gauge groups and topological terms. For continuous gauge groups, there are also codimension-3 't Hooft-Polyakov monopole operators that carry fluxes of the gauge field on the surrounding 2-sphere; they are absent in the finite group gauge theory considered here.

In this section, we will explain how the topological action changes the magnetic defects and their fusion rules and their statistics. Some previously known examples of such phenomena are as follows.

- In (3+1)D, gauge theories with continuous gauge groups can have theta terms [108, 109] (and possibly discrete theta terms [110, 111]). These topological actions change the spectrum of line operators: while the spectrum of Wilson lines is not affected, the spectrum of 't Hooft lines becomes modified with additional "fractional" electric charges that are the projective representation of the gauge group, which is the Witten effect [84]. As a result, the 't Hooft lines become dyons, and their fusion rule is modified. In the absence of topological action, the fusion of 't Hooft lines always produces 't Hooft lines, while in the presence of topological action, they become dyons, and their fusion can produce Wilson lines. In addition, the correlation function of the line operator is also modified: the statistics of the particles also depend on the topological action. For instance, a theta term in $SO(N)$ gauge theory with $\theta = 2\pi$ does not change the spectrum of line operators, but the self-statistics of the basic monopole changes from boson to fermion or vice versa (see *e.g.* Refs. [112, 113, 114, 53]).

- Similar phenomena occur in (2+1)D Chern-Simons theory, where the topological Chern-Simons interaction modifies the 't Hooft lines with the magnetic charge to carry electric charges, which is the flux attachment. The fusion rules and statistics of the 't Hooft lines also depend on the Chern-Simons topological action. For instance, the spin of the monopole operator attached to the Wilson line depends on the topological Chern-Simons term [115, 116, 117, 118, 119, 120, 103].

- It also occurs in higher-form gauge theory. For finite group $n$-form gauge theory, the magnetic defect that carries holonomy of the gauge field has codimension $n + 1$, and in the presence of a topological action for the $n$-form gauge field, it is attached with a codimension-$n$ topological defect that supports a topological action for the $n$-form gauge field. For instance, in 2-form $\mathbb{Z}_N$ gauge theory in (3+1)D, where the magnetic defect is a line operator, the topological action $\frac{Np}{4\pi} \int bb$ for 2-form gauge field $b$ implies that the magnetic line defect attaches to the Wilson surface $p \int b$ (see *e.g.* [36, 53]).

We will see that similar phenomena occur in finite group gauge theories for magnetic defects.

## 2.1 Gauged SPT defects

The main players in the discussion are defects described by topological finite group $G$ gauge theory on the submanifold that support the defect. The insertion of such a defect on an $n$-dimensional submanifold $M_n \subset M_D$ is equivalent to modifying the path integral with an additional weight that depends on the $G$ gauge field $a$

$$e^{i \int_{M_n} \mathcal{L}[a]}, \tag{3}$$

where $\int_{M_n} \mathcal{L}[a]$ is a topological action of the gauge field supported on $n$-dimensional submanifold $M_n$, and the defect is invariant under small deformations of $M_n$ for finite group gauge field $a$, which is flat, and thus the insertion represents a topological defect supported on $M_n$.[1] These defects can be obtained from the following construction; one starts with a $D$-dimensional $G$-SPT phase, characterized by a cocycle $[\omega^{(D)}] \in H^D(BG, U(1))$, and then decorates an $n$-dimensional submanifold with a lower dimensional $G$-SPT phase, characterized by a cocycle $[\eta^{(n)}] \in H^n(BG, U(1))$. Finally, one gauges the $G$ symmetry by summing over all flat $G$-gauge field configurations. These defects generate an Abelian $(D - n - 1)$-form symmetry, with the group multiplication law given by stacking the SPT phases. We will refer to such defects as the gauged SPT defects. For instance, when the submanifold is one-dimensional, $(0+1)$D SPT phases are vacuum electric charges. Such defects are the Wilson lines in the one-dimensional representations of the gauge group $G$, $\int_{M_n} \mathcal{L}[a] = \oint_{M_1} a^* \eta^{(1)}$, with $\eta^{(1)} \in H^1(BG, U(1)) = \text{Hom}(G, U(1))$ labeling the one-dimensional representations of $G$, where $a^* \eta^{(1)}$ is the pullback of $\eta^{(1)}$ by $a$. For other instances of gauged SPT defects, see $e.g.$ Refs. [123, 44, 124, 65, 83].

We believe that all invertible defects of codimension greater than two in Dijkgraaf-Witten theories in spacetime dimension $D > 3$ come from such gauged SPT defects obtained by gauging lower-dimensional $G$-SPT phases supported on submanifolds. For codimension-2, we expect invertible defects come from either Abelian magnetic defects or from $(D - 2)$-dimensional gauged $G$-SPT phases; some evidence for this was given in Ref. [83] by comparing with layer constructions. For codimension-1 and $D > 3$, invertible defects can arise from gauging codimension-1 SPT phases or from elements of $\text{Aut}(G)$; in $D = 3$ there are more exotic examples of invertible codimension-1 defects since electric and magnetic excitations can be permuted [70, 83].

For $n = 1$, the magnetic defects transform linearly under the $(D - 2)$-form symmetry, with eigenvalue given by evaluating the character of the Wilson line with respect to the holonomy carried by the magnetic defect, which is a conjugacy class of $G$. On the other hand, for $n > 1$, there are no objects transformed as linear representations of these higher-form symmetries, but the generators of these higher-form symmetries are nonetheless non-trivial. In particular, these gauged SPT defects participate in non-trivial correlation functions with multiple magnetic defects, as we will discuss in later sections.

## 2.2 Magnetic defects are attached to gauged SPT phases

We will show that the pure magnetic defect in the presence of the topological action carries an anomalous $G$ symmetry. The anomalous $G$ symmetry can be described by a $G$ gauge theory living on a codimension-1 defect whose boundary supports the magnetic defect. Let us consider the codimension-2 magnetic defect supported at the origin of $\mathbb{R}^2$, extending in the remaining $(D - 2)$-directions, and we elongate $\mathbb{R}^2$ into a semi-infinite cigar with holonomy given by the conjugacy class $[g]$ of $g \in G$ supported on the circumference, which is a circle fibration over the radial direction $[0, \infty)$. See Figure 1 for illustration. We then reduce the theory onto the magnetic defect. The presence of the topological action for the $G$ gauge field implies that the magnetic defect is dressed with a gauged SPT phase, given by integrating the topological action along the circle fiber. Denote the partition function by $Z \propto \sum_{[a]} e^{i \int_{\mathbb{R}^2 \times \Sigma_{D-2}} \mathcal{L}[a]}$ for some $(D - 2)$ dimensional submanifold $\Sigma_{D-2}$, and let us take $g \in Z(G)$ to be in the center of the gauge group,

$$Z[\mathbb{R}^2 \times \Sigma_{D-2}, da|_{x \to 0 \in \mathbb{R}^2} \sim g\delta^2(x)] = \frac{1}{|G|} \sum_{[a]: \oint_{S^1(0)} a = g} e^{i \int_{\mathbb{R}^2 \times \Sigma_{D-2}} \mathcal{L}[a]} = \frac{1}{|G|} \sum_{[a]: \oint_{S^1(0)} a = g} e^{i \int_{[0, \infty) \times \Sigma_{D-2}} \mathcal{L}[a, g]} , \quad (4)$$

where $\mathcal{L}[a, g] = \oint_{S^1(0)} \mathcal{L}[a]$ with $S^1(0)$ being a small circle surrounding the origin $0 \in \mathbb{R}^2$ where the magnetic defect is inserted. The additional weight in the above path integral indicates the presence of the gauged SPT defect inserted at $[0, \infty) \times \Sigma_{D-2}$ whose boundary is the magnetic defect. Examples of such methods of deriving the gauged SPT defect attached to the magnetic defect in a higher-form gauge theory with topological action are discussed in $e.g.$ Ref. [53].

In the following, we will focus on the magnetic defects that carry the holonomy of $G$ gauge field that takes value in the center $Z(G)$.

Consider the topological action of $G$ gauge field given by a Dijkgraaf-Witten term $[\omega^{(D)}] \in H^D(BG, U(1))$ with cocycle representative $\omega^{(D)}$. In general, for finite group $G$, the element has finite order $k$, and we can fix a cocycle representative such that it takes value $\frac{2\pi}{k}\mathbb{Z}$ on any $n$ group elements:

$$\omega^{(D)} = \frac{2\pi}{k}(\omega^{(D)})_k \mod 2\pi\mathbb{Z} , \quad (5)$$

---

[1] If the gauge group is continuous instead of finite, for the defect to be topological, one also needs to require the displacement operators to vanish, which are the components of the energy-momentum tensor in the directions normal to the defect [121] (see $e.g.$ Refs. [60, 62, 122] for examples of topological defects in $(3+1)$D Maxwell theory).

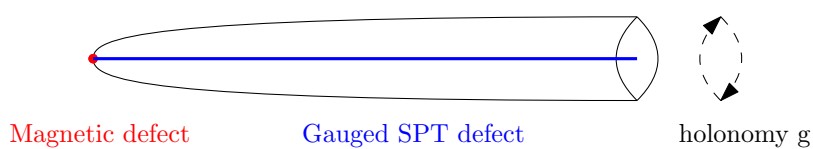

Magnetic defect      Gauged SPT defect      holonomy g

Figure 1: The magnetic defect is attached to a gauged $G$ SPT defect (that supports $G$ gauge theory with a topological action) in the presence of topological action, with the gauged SPT defect given by integrating the topological action over the circle fiber with given holonomy of the magnetic defect. The magnetic and gauged SPT defects both extend in the remaining transverse $(D-2)$ dimensions, which we suppress in the figure.

where the subscript $k$ in $(\omega^{(D)})_k$ means that it is a cocycle with coefficient in $\mathbb{Z}_k$.

For the topological action $S[a] = a^*\omega^{(D)}$, the circle reduction with holonomy $g \in Z(G)$ is described by the degree-$(D-1)$ cocycle given by the slant product of $\omega^{(D)}$ with respect to $g$ [82], $S[a,g] = \int_{S^1} \omega^{(D)} = i_g \omega^{(D)}$. Thus the gauged SPT phase attached to the magnetic defect of conjugacy class $g$ in the center $Z(G)$ is[2]

$$(i_g\omega^{(D)})(g_1,\cdots,g_{D-1}) \equiv \sum_m (-1)^m \omega^{(D)}(g_1,\cdots,g_{m-1},g,g_m,\cdots,g_{D-1}), \quad g_i \in G \ . \tag{7}$$

For finite group $G$, the cocycle $i_g\omega^{(D)}$ evaluated on any $(D-1)$ group elements is a $k$th root of unity. It can be expressed as the map of a $\mathbb{Z}_k$-valued cocycle using the inclusion $\iota : \mathbb{Z}_k \to U(1)$,

$$i_g\omega^{(D)} = \frac{2\pi}{k}\left(i_g\omega^{(D)}\right)_k \mod 2\pi\mathbb{Z} \ , \tag{8}$$

for $\mathbb{Z}_k$ valued cocycle $\left(i_g\omega^{(D)}\right)_k$. If the element $[(i_g\omega^{(D)})]$ in $H^{D-1}(BG, U(1))$ has order $k'$ (which must be a divisor of $k$), it can be represented by a $\mathbb{Z}_{k'}$ cocycle

$$\frac{2\pi}{k'}\left(i_g\omega^{(D)}\right)'_{k'}, \quad [\frac{2\pi}{k'}\left(i_g\omega^{(D)}\right)'_{k'}] = [(i_g\omega^{(D)})] \in H^{D-1}(BG, U(1)) \ . \tag{9}$$

Then we can decompose

$$\left(i_g\omega^{(D)}\right)_k = \frac{k}{k'}\left(i_g\omega^{(D)}\right)'_{k'} + d(i'_g\omega^{(D)})_{k''}/k'' \ , \ , \tag{10}$$

for some $(D-2)$ cocycle denoted by $(i'_g\omega^{(D)})_{k''}$ that takes value in $\mathbb{Z}_{k''}$ for some integer $k''$.

Let us denote the above decomposition of $i_g\omega^{(D)}$ by

$$i_g\omega^{(D)} = i_g^A\omega^{(D)} + \frac{1}{|\omega^{(D)}|}d(i_g^B\omega^{(D)}) \ , \tag{11}$$

where $|\omega^{(D)}| = k$ is the order of $[\omega^{(D)}]$ in $H^D(BG, U(1))$, and $i_g^B\omega^{(D)} \in \frac{2\pi}{k''}\mathbb{Z}$.[3]

The reason that we distinguish these two is that while the first part $(i_g\omega^{(D)})_{k'}$ describes non-trivial defects that support SPT phase with dynamical $G$ gauge field in $(D-1)$ dimension, the second part $(i'_g\omega^{(D)})_{k''}$ does not. Intuitively, it is a fraction $1/k''$ of a defect that supports the SPT phase with dynamical $G$ gauge field in $(D-2)$ dimension. Nevertheless, the second part also plays an important role in the correlation functions of the magnetic defects. The second part can be detected by the trivalent junction of fusing magnetic defects, as in Figure 2. Fusing two magnetic defects that carry holonomies $g, g' \in Z(G)$ produces a magnetic defect of holonomy $g + g' \in Z(G)$, and the junction emits a $(D-2)$-dimensional defect that carries an SPT phase with dynamical gauge field $G$, given by

$$\Omega(g,g') = \frac{1}{|\omega^{(D)}|}\left(i_g^B\omega^{(D)} + i_{g'}^B\omega^{(D)} - i_{g+g'}^B\omega^{(D)}\right) \ . \tag{12}$$

To summarize, the effect of topological interaction $\omega^{(D)}$ of the $G$ gauge fields have two effects on the magnetic defect:

---

[2]One can verify the result of the slant product on a cocycle is also closed,

$$(di_g\omega^{(D)})(g_1,\cdots,g_D) = \sum_{m'}(-1)^{m'}(i_g\omega^{(D)})(g_1,\cdots,g_{m'-1},g_{m'+1},\cdots) = -d\omega^{(D)}(g_1,g,\cdots,g_D) = 0 \ . \tag{6}$$

Properties of slant product and its generalization can be found in *e.g.* [125].

[3]In general, the inclusion map $\mathbb{Z}_k \to U(1)$ induces a map for the cocycles $Z^*(BG, \mathbb{Z}_k) \to Z^*(BG, U(1))$, where some $\mathbb{Z}_k$-valued cocycles become exact $U(1)$-valued cocycles under the inclusion map.

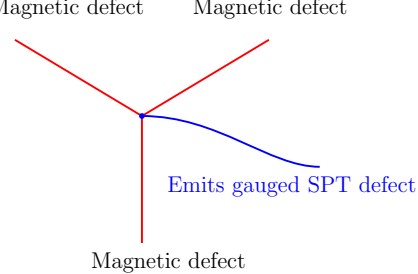

Figure 2: Junction of magnetic defects can emit a gauged SPT defect of dimension $(D-2)$.

(1) The magnetic defect lives on the boundary of a gauged SPT defect in one dimension higher as in Figure 1, which is decorated with $G$ gauge theory with topological action $[i_g^A \omega^{(D)}] = [i_g \omega^{(D)}] \in H^{D-1}(BG, U(1))$ for a magnetic defect with holonomy $g \in Z(G)$.

(2) The trivalent junction of the magnetic defects lives on the boundary of a gauged SPT defect in the same dimension as the magnetic defect as in Figure 2, given by $\Omega(g, g')$ in (12) for the junction of three magnetic defects with holonomies $g, g', g + g' \in Z(G)$.

### 2.2.1 Generalization to continuous gauge groups

Let us reproduce the known results on the Witten effect in gauge theory in (3+1)D, and the flux attachment in Chern-Simons theory in (2+1)D, using the above method.

We note that for a continuous gauge group such as $U(1)$, one can also consider a magnetic monopole (a magnetic defect of codimension 3 in spacetime). Take the spacetime to be locally $\mathbb{R}^3 \times \mathbb{R}^{D-3}$, then the monopole sits at a point $x \in \mathbb{R}^3$ and spans the entire $\mathbb{R}^{D-3}$. The monopole can be surrounded by a sphere $S^2 \times \{\text{point } y\} \subset \mathbb{R}^3 \times \mathbb{R}^{D-3}$, where $S^2 \subset \mathbb{R}^3$ encloses the point $x \in \mathbb{R}^3$; we note that shrinking the size of $S^2$ until the sphere becomes a point intersects the monopole once at a point $(x, y) \in \mathbb{R}^3 \times \mathbb{R}^{D-3}$. The sphere carries magnetic flux $\{m_i\}$ of the Cartan subgroup $U(1)^r$ of the gauge group of rank $r$ [126, 127, 128].

We consider the setup of elongating $\mathbb{R}^3$ to be the cigar geometry of $S^2$ fibers over the radial direction, with flux $\{m_i\}$ supported on $S^2$, $\oint_{S^2} F^i = 2\pi m_i$, where $F^i$ with $i = 1, \cdots, r$ are the field strengths for the Cartan subgroup $U(1)^r$ of the gauge group. By integrating the topological action of the gauge field over the $S^2$ fiber with the prescribed magnetic fluxes, we can deduce the gauged SPT defect attaching to the magnetic monopole sitting at a point in $\mathbb{R}^3$ (and span the remaining $(D-3)$ dimensions).

**The Witten effect**   In (3+1)D spacetime, consider the topological $\theta$ term for $U(1)$ gauge theory,

$$\frac{\theta}{2(2\pi)^2} FF, \quad F = da . \tag{13}$$

Integrating the theta term over the fiber $S^2$ implies that the monopole is attached to

$$\frac{m\theta}{2\pi} \int F , \tag{14}$$

which is the analogue of the total derivative contribution $i_g^B$. Using $F = da$, we find that the monopole with magnetic charge $m$ carries fractional electric charge $m\theta/2\pi$. This is the Witten effect [84].

**Flux attachment**   Similarly, consider $U(1)_k$ Chern-Simons theory in (2+1)D,

$$\frac{k}{4\pi} \int ada . \tag{15}$$

Integration over $S^2$ fiber with flux $\oint da = 2\pi$ can be carried out by decomposing $a = a + 2\pi\tau$, where $\tau$ is the Berry connection on $S^2$ whose field strength has unit flux. This produces

$$k \int a , \tag{16}$$

and thus the basic monopole operator is attached to charge $k$ Wilson line that ends on the point where the monopole operator is located.

## 2.3 Magnetic defects of center-holonomy can have non-Abelian fusion

Let us explain how the topological action modifies the fusion rule of the magnetic defects. The modification of the fusion rule originates from the anomalous $G$ symmetry on the defect that is present due to the topological action of the $G$ gauge field. In other words, the anomalous $G$ symmetry on the magnetic defect arises because the magnetic defect is at the boundary of a gauged SPT phase of one dimension higher. To incorporate such anomalous $G$ symmetry, the magnetic defect that couples to the bulk gauge theory must have extra degrees of freedom, and this changes the fusion rule.

In the following, we will show that even when the holonomy carried by the magnetic defect is in the center of the gauge group, the fusion rule of the magnetic defects can become non-Abelian in the presence of non-trivial topological action of the $G$ gauge field. We will focus on the magnetic defects with holonomies in the center of the gauge group.

### 2.3.1 Fusion of magnetic defect with codimension-2 gauged SPT defect

Due to the anomalous $G$ symmetry on the magnetic defect, under a $G$ gauge transformation by $\lambda$ that takes value in $G$, the magnetic defect changes by an amount given by the anomaly inflow mechanism, which we will call the anomaly descendant.

Denote the anomaly descendant for an $n$ cocycle $\eta^{(n)} \in Z^n(BG, U(1))$ for group $G$ by $j_\lambda(\eta^{(n)}) \in C^{n-1}(BG, U(1))$, for a gauge transformation by a 0-cochain $\lambda$. In a simplicial formulation, we transform the gauge element on link $(01)$ connecting vertices $0,1$ by $g(01) \to \lambda_0^{-1} g(01) \lambda_1$. $j_\lambda$ is defined by the equations:

$$dj_\lambda(\eta^{(n)})(g_1, \cdots, g_n) = \eta^{(n)}(\lambda_1^{-1} g_1 \lambda_2, \cdots, \lambda_n^{-1} g_n \lambda_{n+1}) - \eta^{(n)}(g_1, \cdots, g_n), \quad j_\lambda|_{\lambda=0} = 0 . \tag{17}$$

The solution $j_\lambda(\eta^{(n)})$ in the above equation can be interpreted as the anomalous transformation of the partition function, $Z_{\text{boundary}} \to Z_{\text{boundary}} e^{i \int j_\lambda(\eta^{(n)})}$, on the boundary of the bulk SPT phase with effective action $\eta^{(n)}$, under the background gauge transformation $\lambda$. We would like to obtain the anomalous transformation when the gauge transformation on the boundary is global, $\lambda_i = \lambda$. In such cases, the boundary partition function is multiplied with the partition function of an $(n-1)$-dimensional SPT phase with symmetry given by the centralizer subgroup of $\lambda$ in $G$. For the centralizer subgroup to be the entire $G$, let us take $\lambda_i = \lambda \in Z(G)$ the center of $G$. The corresponding $(n-1)$-dimensional SPT phase with $G$ symmetry can be obtained as follows. Let us consider the bulk geometry to be $[0,1]_x \times M$ for closed $M$ and an interval $0 \le x \le 1$, and we perform gauge transformation by $\lambda \in Z(G)$ on one end $x = 1$ relative to the other end $x = 0$, which can be viewed in the following two equivalent ways:

- This is a global transformation on the boundary $\{x = 1\} \times M$ by parameter $\lambda$, and it produces extra $(n-1)$ dimensional SPT phase with effective action $j_\lambda \eta^{(n)}|_{\lambda_i = \lambda}$.

- This gives the same phase from reducing the effective action $\eta^{(n)}$ for the $n$-dimensional SPT phase on a circle with holonomy $\lambda$, where the circle is obtained by gluing the two ends of the interval $0 \le x \le 1$.

By comparing the two descriptions, we have

$$j_\lambda(\eta^{(n)})|_{\lambda_i = \lambda} = i_\lambda \eta^{(n)}, \quad \lambda \in Z(G) . \tag{18}$$

We thus have the following fusion rule. Denote the magnetic defect by $U_g(M_{D-2})$, and the gauged SPT defect of dimension $n$ by $\mathcal{W}_{\alpha^{(n)}}(M_n)$ for cocycle $\alpha^{(n)} \in Z^n(BG, U(1))$. Then by taking the transformation with constant $\lambda \in Z(G)$, we find the fusion rule

$$U_g \times \mathcal{W}_{i_\lambda i_g \omega^{(D)}} = U_g , \quad \forall \lambda \in Z(G) . \tag{19}$$

The fusion rule is the statement that in the presence of $G$ gauge field, under a $G$ global transformation on the magnetic defect there is an anomalous shift depending on the anomaly of $G$ symmetry. Thus the magnetic defect is non-Abelian if and only if $[i_g \omega^{(D)}]$ is a non-trivial class in $H^{D-1}(BG, U(1))$.

### 2.3.2 Fusion among magnetic defects

Similarly, we can consider fusing two magnetic defects. Then the fusion of two magnetic defects is:

$$U_g \times U_{g'} = U_{g+g'} \frac{1}{\mathcal{N}} \left( \sum_{\lambda \in Z(G)} \mathcal{W}_{i_\lambda i_g \omega^{(D)}} \right) \left( \sum_{\lambda' \in Z(G)} \mathcal{W}_{i_{\lambda'} i_{g'} \omega^{(D)}} \right) / \{ \mathcal{W}_{i_{\lambda''} i_{g+g'}(\omega^{(D)})} : \lambda'' \in Z(G) \} , \tag{20}$$

where the quotient simplifies the result of the fusion outcome using (19), and the normalization factor $\mathcal{N}$ is included such that the trivial fusion channel only appears once, *i.e.* $U_g \times U_{g'} = U_{g+g'} + \cdots$ with $\cdots$ containing gauged SPT defects.

### 2.3.3 Non-Abelian braiding of magnetic defects

We remark that in addition to the above non-Abelian fusion, for the magnetic defects that are attached to non-trivial gauged SPT defect in one dimension higher, *i.e.* the magnetic defects with holonomy $g$ such that $[i_g^A \omega^{(D)}] \neq 0$, the braiding of the magnetic defects becomes non-Abelian (when the spacetime dimension allows such braiding).

Denote the support of the magnetic defect by a $(D-2)$-dimensional submanifold $M_{D-2}$, which is on the boundary of $V_{D-1}$ that supports the gauged SPT defect attached to the magnetic defect, given by $[i_g^A \omega^{(D)}] \in H^{D-1}(BG, U(1))$. When two magnetic defects supported on $(D-2)$-dimensional submanifolds $M_{D-2}, M'_{D-2}$ have non-trivial linking, $M_{D-2}$ intersects with $V'_{D-1}$ whose boundary is $M_{D-2}$, and similarly $M'_{D-2}$ intersects with $V_{D-1}$ whose boundary is $M_{D-2}$. Since $V_{D-1}, V'_{D-1}$ support non-trivial gauged SPT defects, and the intersection of the magnetic defect with the gauged SPT defect produces lower-dimensional gauged SPT defects, the linking of two magnetic defects with holonomies $g, g'$ produces extra gauged SPT defects, given by $\mathcal{W}_{i_g i_{g'} \omega^{(D)}}$ and $\mathcal{W}_{i_{g'} i_g \omega^{(D)}}$. In other words, the braiding of two magnetic excitations does not return to the original configuration, and the braiding becomes non-Abelian.[4]

### 2.3.4 Example of magnetic defect with center holonomy obey non-Abelian fusion

Consider twisted $\mathbb{Z}_2^3$ gauge theory in (2+1)D, with $\mathbb{Z}_2$ gauge fields $a, b, c$, and the action

$$\pi \int (a \cup du + b \cup dv + c \cup dw + a \cup b \cup c) \ , \tag{21}$$

where $u, v, w$ are $Z_2$ cochains that act as Lagrangian multipliers enforcing $a, b, c$ to be cocycles. Integrating out $v, w$ and $a$ gives

$$du = b \cup c \ , \tag{22}$$

which describes the gauge field of the Dihedral group of order 8. Thus the theory is equivalent to untwisted gauge theory with gauge group given by the Dihedral group of order 8. The Wilson line in the two-dimensional representation of the Dihedral group corresponds to

$$W_{\mathbf{2}} = (-1)^{\oint_\gamma u + \int_\Sigma b \cup c} \ , \tag{23}$$

where $\gamma = \partial \Sigma$, and it does not depend on the surface $\Sigma$: suppose we choose a different surface $\Sigma'$ with $\partial \Sigma' = \gamma$, then $\Sigma \cup \overline{\Sigma'} \equiv \Sigma''$ is a closed surface, and the operator changes by

$$(-1)^{\oint_{\Sigma''} (du + b \cup c)} = 1 \ , \tag{24}$$

where we used the equation of motion, and thus the operator $W_{\mathbf{2}}$ does not depend on the bounding surface.[5] This is an example of a magnetic defect, associated with $\mathbb{Z}_2$ flux of $a$, being attached to a higher dimensional gauged SPT, described by $b \cup c$, and thus being endowed with non-Abelian fusion rules.

There are three Wilson lines in one-dimension representations:

$$W_{\mathbf{1}_e} = (-1)^{\oint_\gamma b}, \quad W_{\mathbf{1}_m} = (-1)^{\oint_\gamma c}, \quad W_{\mathbf{1}_f} = (-1)^{\oint_\gamma (b+c)} \ . \tag{25}$$

We can compute their fusion rule as above: [129]

$$\begin{aligned}
&W_{\mathbf{1}_e}^2 = W_{\mathbf{1}_m}^2 = W_{\mathbf{1}_f}^2 = 1, \\
&W_{\mathbf{2}} \times W_{\mathbf{1}_e} = W_{\mathbf{2}}, \quad W_{\mathbf{2}} \times W_{\mathbf{1}_m} = W_{\mathbf{2}}, \quad W_{\mathbf{2}} \times W_{\mathbf{1}_f} = W_{\mathbf{2}}, \\
&W_{\mathbf{2}} \times W_{\mathbf{2}} = 1 + W_{\mathbf{1}_e} + W_{\mathbf{1}_m} + W_{\mathbf{1}_f} \ .
\end{aligned} \tag{26}$$

This agrees with the tensor product decomposition of the representations of Dihedral group of order 8.

Similarly, one can show that other magnetic lines all carry non-trivial projective representation and become non-invertible. Thus the center of gauge group does not give non-trivial invertible 1-form symmetry.

---

[4]Here we are using the terminology "non-Abelian braiding" loosely: the topological defects do not return to their original topological classes after the braid operation, which is reminiscent of $G$-crossed braiding of symmetry defects in (2+1)D [23]. This is not quite the same as the case where braiding induces a unitary transformation on a topologically degenerate subspace.

[5]We note that one cannot set $\int (du + b \cup c) = 0$ on open surfaces, since we need to specify additional boundary condition. Alternatively, the equation of motion is violated when the open surface intersects the line defect $\oint a$; for closed surfaces, such violations sum to zero.

### 2.3.5  Total invertible 1-form symmetry is a central extension

Consider the subset of magnetic defects that are invertible: these are the magnetic defects with holonomy $g \in Z_\omega(G)$ for the subgroup $Z_\omega(G) \subset Z(G)$ given by

$$Z_\omega(G) \equiv \{g \in Z(G): \quad [i_g \omega^{(D)}] = 0 \in H^{D-1}(BG, U(1))\} \ . \tag{27}$$

Since the trivalent juction of magnetic defects that describe their fusion can emit extra gauged SPT defects in $H^{D-2}(BG, U(1))$, the 1-form symmetry $\mathcal{G}^{(1)}$ generated by the codimension-2 invertible magnetic defects and the gauged SPT defect of dimension $(D-2)$ form a group extension

$$1 \to H^{D-2}(BG, U(1)) \to \mathcal{G}^{(1)} \to Z_\omega(G) \to 1 \ . \tag{28}$$

The extension is specified by a 2-cocycle $\Omega$ that depends on two elements of $Z_\omega(G)$ and takes value in the Abelian group $H^{D-2}(BG, U(1))$. For $g, g' \in Z_\omega(G)$, it is given by

$$[\Omega(g, g')] = \left[(i_g^B \omega^{(D)} + i_{g'}^B \omega^{(D)} - i_{g+g'}^B \omega^{(D)})/|\omega^{(D)}|\right] \in H^{D-2}(BG, U(1)) \ . \tag{29}$$

We will refer the background gauge field for the 1-form center symmetry generated by the magnetic defect as $B_2$. Let us denote the background for the 1-form symmetry generated by the gauged SPT defects in $H^{D-2}(BG, U(1))$ as $C_2$. (Similarly, we will denote the background for the $n$-form symmetry $H^{D-n-1}(BG, U(1))$ generated by the gauged SPT defects by $C_{n+1}$.) Then in the absence of other background gauge fields, the background gauge fields for the 1-form symmetry obey the relation

$$dC_2 = \text{Bock}(B_2) \ , \tag{30}$$

where Bock is the Bockstein homomorphism for (28).

## 2.4  Correlation function of magnetic defects

### 2.4.1  Self-statistics

The magnetic defects have non-trivial correlation function due to the attached gauged SPT defect. Let us compute the correlation function, which gives the self-statistics of the magnetic defect.

Denote the support of the magnetic defect by $M_{D-2}$, which is on the boundary of $V_{D-1}$. If the magnetic defect carries a holonomy $g$, it has the effect of sourcing a background contribution to the gauge field[6]

$$a = a_g, \quad a_g \equiv g\delta(V_{D-1})^\perp \ , \tag{31}$$

where $\delta(V_{D-1})^\perp$ is the delta function 1-form that restricts to $V_{D-1}$, and it is the Poincaré dual 1-form of $V_{D-1}$. The correlation function of the magnetic defect itself is

$$\langle U_g(M_{D-2})\rangle = \exp\left(i \int_{V_{D-1}} (a_g)^*(i_g \omega^{(D)})\right) \ , \tag{32}$$

where $(a_g)^*(i_g \omega^{(D)})$ is the pullback of the cocycle to spacetime by the gauge field configuration $a = a_g$. The correlation function will in general depend on the framing of $M_{D-2}$.

**Example: semion in twisted $\mathbb{Z}_2$ gauge theory in (2+1)D**  Consider $\mathbb{Z}_2$ with topological action given by the non-trivial element $\omega$ in $H^3(B\mathbb{Z}_2, U(1)) = \mathbb{Z}_2$. Then for the non-trivial element $g = 1 \in \{0, 1\} = \mathbb{Z}_2$, the pullback of $i_1(\omega)$ by the $\mathbb{Z}_2$ gauge field $a$ is $\pi da/2$. Then the self-statistics of the magnetic defect is described by the correlation function

$$\langle U_g(M_1)\rangle = \exp\left(\frac{\pi i}{2} \int_{V_2} d\delta(V_2)^\perp\right) = \exp\left(\frac{\pi i}{2} \int \delta(V_2)^\perp d\delta(V_2)^\perp\right) = i^{\#(M_1, V_2)} \ , \tag{33}$$

where $\#(M_1, V_2)$ is the self linking number of $M_1$. This reproduces the semion self-statistics $i$ for the magnetic defect.

---

[6]In other words, we set $a = a_g + a'$ with fluctuating $a'$ of zero flux. The correlation can be evaluated locally near the defects and thus we can take the spacetime to be a $D$-dimensional sphere, then $a'$ is trivial.

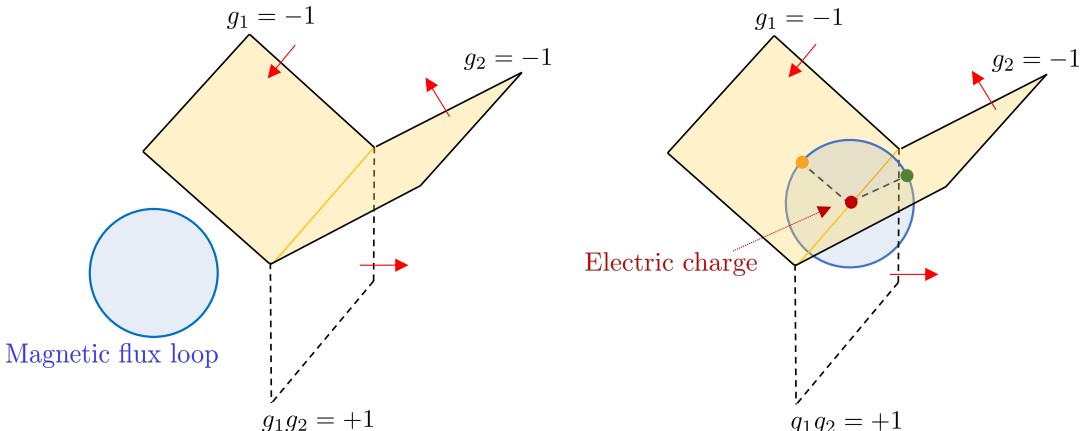

Figure 3: Junction of 0-form symmetry defects indicated by black and dashed lines (with red arrows indicating their orientations to specify the action of the symmetry). On the right, we include a magnetic flux loop encircling the junction, and there are semion and anti-semion particles at the intersection of the flux loop with the non-trivial domain walls, indicated by the green and orange points. To compensate for their difference, there is an electric charge at the junction, indicated by the red dot.

### 2.4.2 Correlation function of magnetic defects and gauged SPT defects

Similarly, the correlation function in the presence of other gauged SPT defect $\mathcal{W}_{\eta^{(n)}}(M_n)$ is given by

$$\langle U_g(M_{D-2})\mathcal{W}_{\eta^{(n)}}(M_n)\rangle = \exp\left(i\int_{V_{D-1}}(a_g)^*(i_g\omega^{(D)})\right)\exp\left(i\int_{M_n}(a_g)^*(\eta^{(n)})\right) . \tag{34}$$

## 3 Higher-group symmetry in gauge theory with bosons

In this section, we will use examples to illustrate that the symmetry generated by the magnetic defects $U_g$ of codimension-2 mixes with other symmetries generated by gauged SPT defect $\mathcal{W}_{\eta^{(n)}}$ to form a higher-group symmetry. On the other hand, the symmetries generated only by the gauged SPT defects do not mix with each other.

We will demonstrate such a mixing between symmetries by examining the configuration of magnetic defects that intersect with the gauged SPT defect, which creates a magnetic source on the gauged SPT defect. Then the generalized Witten effect implies an additional gauged SPT defect emitted from the intersection. We will demonstrate such a phenomenon using the background gauge fields and the property of gauged SPT phases, and using operator commutation relation in the lattice model.

### 3.1 Example: $\mathbb{Z}_2$ gauge theory in (3+1)D

Let us consider the following symmetries: the 0-form symmetry generated by a domain wall that is decorated with a Dijkgraaf-Witten $\mathbb{Z}_2$ gauge theory, the 1-form symmetry generated by the magnetic surface defect, the 2-form symmetry generated by the Wilson line. These symmetries mix into a 3-group, as discussed in Ref. [44]. In Appendix B we provide another derivation of such 3-group symmetry by embedding the discrete gauge fields into continuous $U(1)$ gauge fields. The 3-group symmetry can be summarized using the following relation between the background gauge fields $C_1, B_2, C_3$ for the $\mathbb{Z}_2$ 0-form, 1-form, and 2-form symmetries:

$$dC_1 = 0, \quad dB_2 = 0, \quad dC_3 = B_2 \cup \frac{d\tilde{C}_1}{2} , \tag{35}$$

where the above equations hold modulo 2. $\tilde{C}_1$ is a lift of $C_1$ to $\mathbb{Z}_4$ coefficients that satisfies $\tilde{C}_1 \bmod 2 = C_1$. Physically, $C_1$ determines whether a domain wall exists, and the lift $\tilde{C}_1$ determines an orientation on the domain wall.[7]

---

[7]The cocycle $d\tilde{C}_1/2$ is the image of the Bockstein homomorphism on $C_1$ [130]. Changing the $\mathbb{Z}_4$ lift by $\tilde{C}_1 \to \tilde{C}_1 + 2c$ with $\mathbb{Z}_2$-valued 1-cochain $c$ amounts to redefinition of background gauge field $C_3 \to C_3 + B_2 \cup c$. Such a redefinition of the background gauge field is commented on in Section 6.

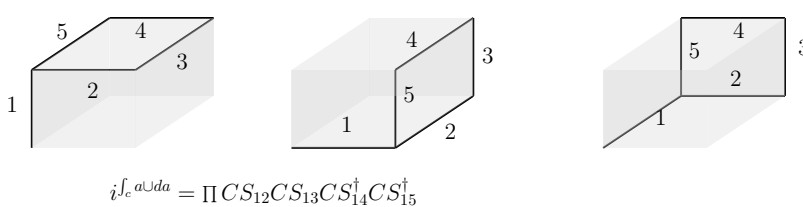

$$i^{\int_c a \cup da} = \prod CS_{12} CS_{13} CS_{14}^\dagger CS_{15}^\dagger$$

Figure 4: The operator $U$ on the cubic in grey shade is given by the product of the control-$S$ gates on the pairs of edges indicated in the figure, where the control-$S$ on edge $1, 2$ with variables $a = a_1, a_2$ of $Z$ eigenvalues in $\{0, 1\} \subset \mathbb{Z}_4$ is given by $i^{a_1 a_2}$.

The last equation can be understood as follows. To create the $\mathbb{Z}_2$ domain wall requires a choice of orientation (see lattice description below). The physical implication of the choice of orientation is that it determines whether a magnetic flux piercing the domain wall induces a semion or an anti-semion.

Now we can consider the junction between two $\mathbb{Z}_2$ domain walls created with opposite orientations. A magnetic flux loop circling the junction induces a semion in one of the domain walls and an anti-semion in the other domain wall, which leaves the boson, which is the $\mathbb{Z}_2$ charge, at the junction (see Figure 3). In other words, the junction emits an extra electric charge in the presence of magnetic loop excitation piercing the domain walls.

The equation $dC_3 = B_2 \cup d\tilde{C}_1/2$ describes 3-group symmetry, with Postnikov class

$$f_4(C_1, B_2) = B_2 \cup d\tilde{C}_1/2 \ . \tag{36}$$

The equation also implies that under a transformation $B_2 \to B_2 + d\lambda$, $C_3 \to C_3 + \lambda d\tilde{C}_1/2$. In other words, if we insert a magnetic defect by performing a 1-form transformation with parameter $\lambda$, then it produces an extra Wilson line. Such mixing of the background gauge transformation is the defining signature of a higher-group symmetry[8], see *e.g.* Refs. [39, 40, 41].

We remark that such mixing (35) has implications for the symmetry fractionalization when the theory is enriched with a unitary $\mathbb{Z}_2$ symmetry [132], as discussed in [44].

In the following, we will provide a description of the structure (35) using the toric code lattice model.

### 3.1.1 Lattice description of 3-group

We can describe $\mathbb{Z}_2$ gauge theory using the (3+1)D toric code model [70]. Let us consider a Euclidean cubic lattice in three spatial dimensions, with a qubit on each edge acted on by the Pauli operators $X, Y, Z$. The Hamiltonian is

$$H = -\sum_v \prod_{e \supset v} X_e - \sum_f \prod_{e \subset f} Z_e \ , \tag{37}$$

where the product of $X$ is over the six edges that meet at the vertex $v$, and the product of $Z$ is over the four edges that surround the face $f$. The $\mathbb{Z}_2$ gauge field is represented by the operator-valued 1-form $a = (1 - Z)/2$ that acts on the qubit on each edge $e$ by $Z_e = (-1)^{a(e)}$.

The symmetries that participate in the 3-group as described in Eq. (35) are generated by the following operators:

- The $\mathbb{Z}_2$ 0-form symmetry is emergent because it keeps the ground state subspace invariant, but does not commute with the full Hamiltonian. It is generated by an operator supported on a three-dimensional region $R$:

$$U(R, \sigma) = \prod_{c \in R} U(c)^{\sigma(c)}, \quad U(c) \, |\{a\}\rangle = (-1)^{\int_c a \cup \frac{d\tilde{a}}{2}} \, |\{a\}\rangle = \prod CS^\sigma \, |\{a\}\rangle \ . \tag{38}$$

Here $\tilde{a}$ denotes the $\mathbb{Z}_4$ lift of $a$ where the non-trivial element of $\mathbb{Z}_2$ is lifted to the generator $1 \in \mathbb{Z}_4$. $S = \sqrt{Z} = \text{diag}(1, i)$ in the eigenbasis of $Z$. Here $\sigma(c) = \pm 1$ is an arbitrary choice of orientation for each cube $c$. The orientation-reversing domain walls where $\sigma(c)$ changes sign determines a 2d surface in space, which will play an important role below. $|\{a\}\rangle$ is the state of the Hilbert space specified by the $\mathbb{Z}_2$ gauge field configuration $\{a\}$ on edges, and $CS$ is the control-$S$ operator with the product over all pairs of edges that have non-trivial contributions to $a \cup d\tilde{a}/2$ on a cube $c$, see Figure 4 for an illustration.

---

[8]When the background fields are dynamical, such a transformation is the analog of the Green-Schwarz mechanism in String theory [131].

The operator supported on the entire space commutes with the Hamiltonian on the low energy subspace with zero flux (while there can be electric charges). Indeed, the commutation relation between $U$ on the entire space and the vertex term of the Hamiltonian on a vertex $v$ in the low energy subspace where $da = 0$ can be computed as [9]

$$\left( \prod_{v \subset e} X_e \right)^{-1} U \left( \prod_{v \subset e} X_e \right) |\{a\}\rangle = (-1)^{\int (a+d\hat{v}) \frac{d(\widetilde{a+d\hat{v}})}{2}} |\{a\}\rangle = (-1)^{\int a \cup \frac{d\tilde{a}}{2} + \text{coboundary}} |\{a\}\rangle$$

$$= U |\{a\}\rangle$$

(39)

where the integral is over the whole space, and $\hat{v}$ is a $\mathbb{Z}_2$ 0-cochain which is nonzero only on the vertex $v$.

We note that in the low energy subspace without the magnetic flux, the operator $U$ is topological, even in the presence of electric charges

If the region $R$ has a non-zero boundary $\partial R$, the operator $U(R, \sigma)$ creates a $\mathbb{Z}_2$ domain wall on $\partial R$, associated with the orientation $\sigma$ restricted to $\partial R$.

- $\mathbb{Z}_2$ 1-form symmetry generated by the membrane operator that is supported on a surface on the dual lattice, given by

$$V(\Sigma) = \prod_{e \cap \Sigma \neq 0} X_e \ , \tag{40}$$

where the product is over the edges that intersect the membrane on the dual lattice. This operator $V(\Sigma)$ supported on closed membrane commutes with the Hamiltonian in the whole Hilbert space, so it is an exact symmetry rather than an emergent symmetry. Note that $V(\Sigma)$ becomes fully topological and generates $\mathbb{Z}_2$ 1-form symmetry only in the subspace where electric particles are absent.

- $\mathbb{Z}_2$ 2-form symmetry generated by the line operator that is supported on a curve on the lattice,

$$W(\gamma) = \prod_{e \in \gamma} Z_e \ , \tag{41}$$

where the product is over the edges on the curve $\gamma$. The line operator $W(\gamma)$ is also an exact symmetry of the Hamiltonian, though it is not fully topological in the presence of magnetic flux excitations. In the low energy subspace where the magnetic excitations are absent, $W(\gamma)$ becomes topological and regarded as a generator of $\mathbb{Z}_2$ 2-form symmetry.

**3-group symmetry as commutation relation** In the following, we will show that these operators obey the following commutation relation on the low energy subspace with zero flux

$$\text{3-group commutation relation} \qquad [U(R, \sigma), V(\Sigma)] \equiv U(R, \sigma) V(\Sigma) U(R, \sigma)^{-1} V(\Sigma)^{-1} = W(\gamma) \ . \tag{42}$$

Here $\gamma$ is the 1d curve along which the orientation-reversing domain walls defined by $\sigma$ intersect the surface $\Sigma$. Figure 5 shows an example where there is a single orientation-reversing domain wall separating two half-spaces.

Let us first show that such a commutation relation is equivalent to the spacetime description of the symmetry defects that we discussed earlier. The orientation-reversing domain wall defined by $\sigma$ in $U$ can be regarded as a trivalent junction of 0-form symmetry generators in spacetime described in Figure 3, where the junction emits a trivial codimension-1 defect. The commutation relation Eq. (42) can then be understood as a result of crossing the magnetic surface operator $V(\Sigma)$ through the junction of 0-form symmetry generators. That is, the commutation relation is regarded as a sequence of operators $U, V, U^{-1}, V^{-1}$ inserted in spacetime at distinct spatial slices, and it can be evaluated by passing the magnetic surface operator $V(\Sigma)$ through $U(R, \sigma)$ which amounts to changing the order of operators. This process involves crossing the worldsheet of the magnetic flux loop through the junction of 0-form symmetries, and leaves the worldline $W(\gamma)$ of the electric charge at the intersection between the junction and the surface $\Sigma$, due to the crossing effect described in Figure 3.

---

[9]In the computation, we use the property $\widetilde{a + d\hat{v}} = \tilde{a} + \widetilde{d\hat{v}} + 2a \cup_1 d\hat{v} \mod 4$, and

$$(a + d\hat{v}) \frac{d(\widetilde{a + d\hat{v}})}{2} = (a + d\hat{v})(\frac{d\tilde{a}}{2} + d(a \cup_1 d\hat{v})) = a \frac{d\tilde{a}}{2} + d(a(a \cup_1 d\hat{v}) + \hat{v} \frac{d\tilde{a}}{2} + \hat{v} d(a \cup_1 d\hat{v})).$$

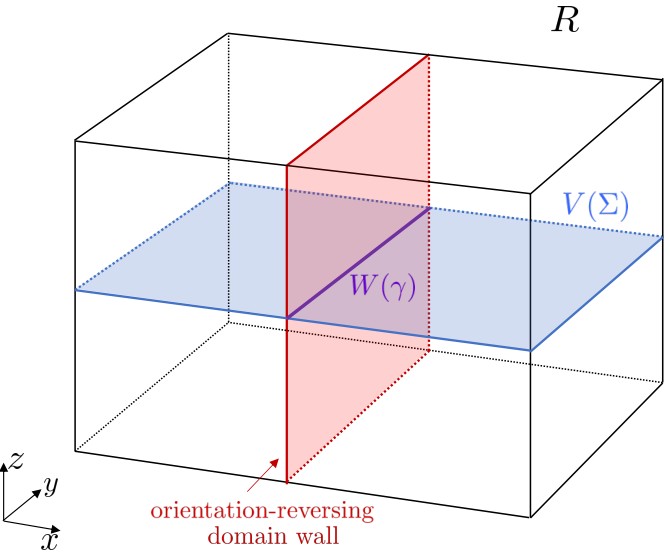

Figure 5: The configurations of the operators $U(R), V(\Sigma), W(\gamma)$ that appear in the commutation relation Eq. (42).

We note that the commutation produces a non-trivial operator that acts on the Hilbert space, instead of a number. Such a commutation relation by definition cannot be viewed as an 't Hooft anomaly and implies that the symmetry forms a higher group structure [40, 41].

We proceed to prove the commutation relation (42). Let us illustrate this commutation relation in a simple setting where we take $\sigma = -1$ on the half-space $x \leq 0$ and $\sigma = +1$ on the half-space $x \geq 0$. The orientation-reversing domain wall is at $x = 0$, and it intersects the membrane operator $V$ placed at $z = 0$, at the line $x = 0, z = 0$ along the $y$ direction. See Figure 5. The membrane operator acts on the collection of edges in the $z$ direction that intersects the entire membrane on the $xy$ plane. Let us denote $\lambda$ to be a $\mathbb{Z}_2$-valued 1-cocycle that takes the nonzero value on edges intersecting the membrane operator $V$, otherwise zero. $\lambda$ is closed $d\lambda = 0$, and it satisfies $\lambda \cup \lambda = 0$. Using $XZX = -Z$, we have

$$VUV^{-1}U^{-1} = (-1)^{\int (a+\lambda)\frac{d(\widetilde{a+\lambda})}{2} - a\frac{d\tilde{a}}{2}} = (-1)^{\int a \cup d(a \cup_1 \lambda) + \lambda \frac{d\tilde{a}}{2}} , \tag{43}$$

where we used $\widetilde{a+\lambda} = \tilde{a} + \tilde{\lambda} + 2a \cup_1 \lambda$ as a $\mathbb{Z}_4$-valued cochain, and also $d\tilde{\lambda}/2 = \lambda \cup \lambda = 0$ mod 2. On the low energy subspace of zero flux $|a\rangle: da = 0$, the term $a \cup d(a \cup_1 \lambda)$ becomes a coboundary and can be ignored. We thus obtain

$$VUV^{-1}U^{-1} = (-1)^{\int \lambda \frac{d\tilde{a}}{2}} = (-1)^{\frac{1}{2}\int d(\tilde{a} \cup \tilde{\lambda})} = (-1)^{\int_{x=0} a \cup \lambda} = (-1)^{\int_{x,z=0} a} = \prod_{e \in y\text{-axis}} Z_e , \tag{44}$$

where we used the fact that the first integral is a total derivative since $(d\tilde{\lambda})/2 = \lambda \cup \lambda = 0$ mod 2, and it can be written as the boundary contribution from the half-space $x \geq 0$ and $x \leq 0$. If there is no reversal of orientation, the two contributions would have been canceled, since the normal direction is opposite on the interface with respect to the two half-spaces. However, since the orientation is reversed on $x \geq 0$, the two contributions add up to $\int_{x=0} a \cup \lambda$, and from the definition of $\lambda$ we find the commutation relation gives $W$ supported on the intersection of the membrane and the domain wall, which is the $y$ axis.

From a similar manipulation, one can show the commutation relation holds for the operators supported on general geometry, where $\lambda$ is some 1-cocycle which becomes nonzero for all edges intersecting the membrane, otherwise zero. For membranes that do not self-intersect, $\lambda \cup \lambda = 0$, the commutation relation gives

$$VUV^{-1}U^{-1} = (-1)^{\int_{\text{domain wall}} a \cup \lambda} = (-1)^{\int_\gamma a} , \tag{45}$$

where the curve $\gamma$ is the intersection of the domain wall with the Poincaré dual of $\lambda$, i.e. the membrane that supports $V$.

When the membrane self-intersects, $\lambda \cup \lambda = d\tilde{\lambda}/2 \neq 0$, and the commutation relation instead has sign $(-1)^{\int \frac{1}{2}d(\widetilde{a \cup \lambda}) + \lambda \frac{d\tilde{\lambda}}{2}}$ on the low energy subspace with zero flux $da = 0$. The additional sign $(-1)^{\int \lambda \frac{d\tilde{\lambda}}{2}}$ can be interpreted as a triple intersection number of the membrane inside the space $(-1)^{\int \lambda \frac{d\tilde{\lambda}}{2}} = (-1)^{\#(\Sigma, \Sigma, \Sigma)}$. [10] This extra sign represents an 't Hooft anomaly of the 3-group symmetry, which we will discuss in more detail in later sections.

---

[10] The triple self-intersection number can be computed by pushing $\Sigma$ off itself using a choice of framing to obtain $\Sigma'$, and then obtaining the intersecting loop $l$. Then we push $\Sigma'$ off itself to get $\Sigma''$ and compute the intersection number of $\Sigma''$ with $l$.

We remark that since the cup product can be defined for any triangulated manifolds, we can also generalize the discussion to the toric code model on any 3d triangulated manifold $M_3$, with a qubit on each edge, and in the Hamiltonian, the vertex term is the product of $X$ on the edges that meet at the given vertex, while the plaquette term is the product of $Z$ on the edges on the boundary of each triangular face. The ground state degeneracy is given by $2^{|H^1(M_3,\mathbb{Z}_2)|}$. We note that $M_3$ does not have to be orientable. The operators $U, V, W$ can be defined in a similar way, and on the low energy subspace with zero flux $da = 0$ they obey the same commutation relation.

### 3.1.2 Breaking the 3-group symmetry

In the toric code Hamiltonian, we can include a transverse field with coupling $J$:[11]

$$H_{\text{transverse}} = -\sum_v \prod X_e - \kappa \sum_f \prod Z_e - J \sum_e X_e \ . \tag{46}$$

Such a term breaks the 2-form symmetry generated by closed Wilson loop $W$ and also breaks the 0-form symmetry generated by $U$. This is consistent with the property that these symmetries mix into a 3-group, which implies that we cannot break the 2-form symmetry while preserving the 0-form and 1-form symmetry. Note that as long as $J$ is small so that we do not pass through a phase transition, the 2-form and 0-form symmetries will both become emergent symmetries.

Instead of $\sum_e X_e$, one can also modify the Hamiltonian with $\sum_e Z_e$, then this breaks the 1-form symmetry generated by the magnetic defect $V$, while preserving the other symmetries, again compatible with the 3-group symmetry structure.

## 3.2 Example: $\mathbb{Z}_2^2$ gauge theory in (3+1)D

Another example is $G = \mathbb{Z}_2 \times \mathbb{Z}_2$ gauge theory in (3+1)D with trivial topological action for the $G$ gauge field. The theory has 0-form symmetry $\mathcal{G}^{(0)} = H^3(BG, U(1)) = \mathbb{Z}_2^3$,[12] 1-form symmetry $H^2(BG, U(1)) \times Z(G) = \mathbb{Z}_2^3$ generated by a gauged SPT defect and the magnetic defects, and 2-form symmetry $H^1(BG, U(1)) = \mathbb{Z}_2^2$. Denote the $\mathbb{Z}_2 \times \mathbb{Z}_2$ gauge field by $a, a'$, then the generators can be written as:

0-form symmetry: $\quad U(R) = (-1)^{\int_R \frac{d\tilde{a}}{2} \cup a'} \ , \quad U'(R) = (-1)^{\int_R \frac{d\tilde{a}}{2} \cup a} \ , \quad U''(R) = (-1)^{\int_R \frac{d\tilde{a}'}{2} \cup a'},$

1-form symmetry: $\quad T(\Sigma) = (-1)^{\int_\Sigma a \cup a'}, \quad \text{magnetic defects } V, V',$

2-form symmetry: $\quad W(\gamma) = (-1)^{\int_\gamma a}, \quad W'(\gamma') = (-1)^{\int_{\gamma'} a'} \ . \tag{47}$

As discussed in [44], the above gauged SPT defects together with 1-form, 2-form symmetry generated by magnetic surfaces and electric Wilson lines form a non-trivial 3-group. In Appendix B we provide another derivation of such 3-group symmetry by embedding the discrete gauge fields into continuous $U(1)$ gauge fields. Such 3-group symmetry can be expressed in terms of their $\mathbb{Z}_2$ background gauge fields. Let us write the background gauge fields of the above gauged SPT defects as $C_1, C_1', C_1'', C_2$, and the background gauge fields for 1-form and 2-form symmetry generated by magnetic and electric operators as $B_2, B_2', C_3, C_3'$, respectively. Following the argument of Appendix B and (3.21) of [44], we find that these background fields satisfy

$$dC_1 = 0, \quad dC_1' = 0, \quad dC_1'' = 0, \quad dC_2 = 0, \quad dB_2 = 0, \quad dB_2' = 0,$$

$$dC_3 = B_2' \cup C_2 + B_2' \cup \frac{d\tilde{C}_1}{2} + \frac{d\tilde{B}_2'}{2} \cup C_1 + B_2 \cup \frac{d\tilde{C}_1'}{2} \ ,$$

$$dC_3' = B_2 \cup C_2 + \frac{d\tilde{B}_2}{2} \cup C_1 + B_2' \cup \frac{d\tilde{C}_1''}{2} \ , \tag{48}$$

where the above equations hold modulo two, and $\tilde{C}_1, \tilde{C}_1', \tilde{C}_1'', \tilde{B}_2, \tilde{B}_2'$ denote some lifts of the $\mathbb{Z}_2$ background gauge fields to $\mathbb{Z}_4$ such that their mod 2 reductions equal $C_1, C_1', C_1'', B_2, B_2'$, respectively. The background gauge fields with the above relation describe a 3-group symmetry.

We note that we can set to zero any of the backgrounds in $C_2, C_1, C_1', C_1''$ in the above equation, and they describe different subgroups of the 3-group. Let us focus on $C_2 = 0$, and $C_1', C_1'' = 0$. The non-trivial 3-group

---

[11]The theory at coupling $J = \kappa$ in fact has a non-invertible symmetry generated by the Kramers-Wannier duality defect, which constrains the low energy dynamics to be non-trivial, as discussed in [60].

[12]There is also a $\text{Aut}(\mathbb{Z}_2 \times \mathbb{Z}_2) = GL(2, \mathbb{Z}_2) = S_3$ 0-form symmetry that permutes the three non-trivial elements in the gauge group $\mathbb{Z}_2 \times \mathbb{Z}_2$, which we do not consider here.

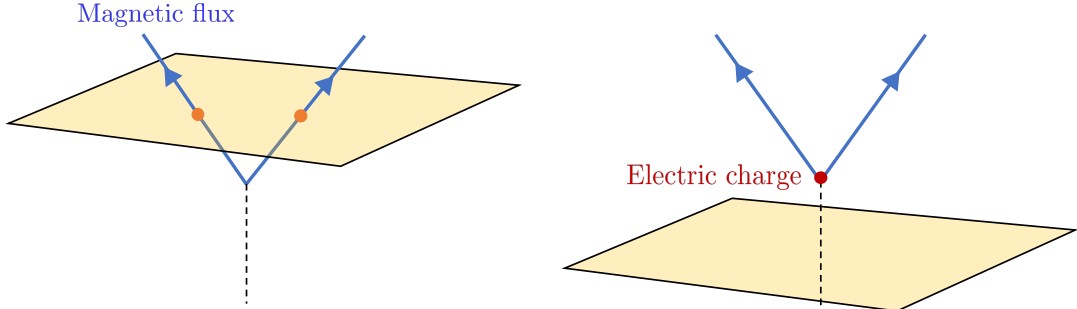

Figure 6: When the magnetic surface has a non-trivial trivalent junction, it is regarded as a fusion of two magnetic flux loops. The magnetic flux at the domain wall is dressed with electric charge $1/2$, which is represented by an orange dot. When the junction crosses through the codimension-1 defect, it leaves an electric charge on the defect due to the $\mathbb{Z}_4$ fusion rule of the magnetic flux on the defect. A similar figure can also be found in [44].

structure involving $C_1, C_3, B_2, B_2'$ are consequences of the following property of the magnetic particles on the domain wall arising from the intersection of the magnetic flux loop with the domain wall:[13]

- The charge-flux attachment implies that the magnetic particle on the domain wall for the gauge field $a$ is attached to electric charge $1/2$ for gauge field $a'$, and the magnetic particle on the domain wall for the gauge field $a'$ is attached to electric charge $1/2$ for the gauge field $a$. This implies that the magnetic particles on the domain wall obey the $\mathbb{Z}_4$ fusion rule, where the fusion of two magnetic particles for the gauge field $a$ (resp. $a'$) gives an electric particle for the gauge field $a'$ (resp. $a$), see Figure 6. This reflects that the (2+1)D $\mathbb{Z}_2 \times \mathbb{Z}_2$ gauge theory with topological action $\frac{d\tilde{a}}{2} \cup a'$ is described by untwisted $\mathbb{Z}_4$ gauge theory, and the magnetic particles behave as the distinct anyons of the untwisted $\mathbb{Z}_4$ gauge theory generating a $\mathbb{Z}_4$ group under fusion.

- In addition, under the orientation reversal of the domain wall, the magnetic particle for $a'$ is dressed with an additional electric particle of $a$, while the magnetic particle for $a$ is left invariant under the orientation reversal. This effect can be understood from the fact that the mutual braiding phases must be complex conjugated under orientation reversal. If we denote the electric and magnetic particles of the (2+1)D $\mathbb{Z}_4$ gauge theory as $E$ and $M$, then the magnetic particles for $a$ and $a'$ can be associated with $E$ and $M$ respectively, while the charge under $a$ and $a'$ is $M^2$ and $E^2$ respectively. Under orientation reversal, we must have $M \to \overline{M} = M \times M^2$, $E \to E$ (or vice versa), which shows that the magnetic particle for $a$ must get attached to a charge of $a'$ (or vice versa). This leads to the asymmetric action of the orientation reversal on the magnetic particles of $a$ and $a'$ as described above.

The attachment of electric charge $1/2$ implies that when the magnetic surface has a non-trivial trivalent junction on the domain wall, it is regarded as a fusion of two magnetic particles on the domain wall, and there is an additional electric charge emitted from the intersection point (see Figure 6). This is the contribution of $\frac{d\tilde{B}_2}{2} \cup C_1$ to $dC_3'$, and the similar contribution of $\frac{d\tilde{B}_2'}{2} \cup C_1$ to $dC_3$. In addition, at the junction of fusing two domain walls into the trivial domain wall, the non-trivial domain walls can meet at the junction with the opposite orientation, and the magnetic defect for the second $\mathbb{Z}_2$ gauge group intersects the junction at a point that emits extra electric charge of the first $\mathbb{Z}_2$ gauge group (this is analogous to the configuration described in Figure 3). This is the contribution of $B_2' \cup \frac{d\tilde{C}_1}{2}$ to $dC_3$.

In the following, we will provide a description of such a 3-group symmetry using the toric code lattice model.

### 3.2.1 Lattice description of 3-group

We consider a three-dimensional euclidean lattice, with two types of qubits on each edge, acted on by the Pauli operators $X, Y, Z$ and $X', Y', Z'$, respectively. The two $\mathbb{Z}_2$ gauge fields are $a = (1 - Z)/2$ and $a' = (1 - Z')/2$. The

---

[13]Another way to see these properties is as follows: we can describe the domain wall theory using $U(1)$ 1-form gauge fields $a, b, a', b'$. The domain wall theory is

$$\frac{1}{2\pi} a' da + \frac{2}{2\pi} adb + \frac{2}{2\pi} a' db' . \tag{49}$$

Then under orientation reversal, $b \to -b, b' \to -b' - a$.

Hamiltonian is two copies of the $\mathbb{Z}_2$ toric codes

$$H^{\text{total}} = H + H' = \left( -\sum_v \prod_{e \supset v} X_e - \sum_f \prod_{e \subset f} Z_e \right) + \left( -\sum_v \prod_{e \supset v} X'_e - \sum_f \prod_{e \subset f} Z'_e \right) , \tag{50}$$

where $H, H'$ are given in the previous example, and in $H'$ the Pauli operators are replaced by the operators with a prime, indicating that they act on the other type of qubit.

We will consider the following symmetry generators:

- 0-form symmetry, generated by

$$U(R) = (-1)^{\int_R \frac{d\tilde{a}}{2} \cup a'} . \tag{51}$$

  The operator supported on the entire space commutes with the Hamiltonian on the low energy subspace with zero flux. The commutation relation between $U$ on the entire space and the vertex term of the Hamiltonian on a vertex $v$ in the low energy subspace where $da = 0, da' = 0$ can be computed as

$$\left( \prod_{v \subset e} X_e \right)^{-1} U \left( \prod_{v \subset e} X_e \right) |\{a\}\rangle = (-1)^{\int \frac{d(\widetilde{a + d\hat{v}})}{2} a'} |\{a\}\rangle = (-1)^{\int \frac{d\tilde{a}}{2} a' + \text{coboundary}} |\{a\}\rangle \tag{52}$$

$$= U |\{a\}\rangle$$

  where the integral is over the whole space, and $\hat{v}$ is a $\mathbb{Z}_2$ 0-cochain which is nonzero only on the vertex $v$. The same relation also holds for the vertex term of the Hamiltonian involving $X'$ operators.

- The 1-form symmetry generated by the membrane operator $V = \prod_e X_e, V' = \prod_e X'_e$, where the two products are over the edges that intersect the membranes on the dual lattice that support the operators $V, V'$, respectively.

- The 1-form symmetry generated by the membrane operator defined on a closed surface $\Sigma$

$$T(\Sigma) = (-1)^{\int_\Sigma a \cup a'} . \tag{53}$$

  The excitation created by this operator is referred to as the twist string in [83].

- The 2-form symmetry generated by the line operators $W = \prod_e Z_e, W' = \prod_e Z'_e$, where the two products are over the edges on the curves that support the operator $W, W'$, respectively.

We will show that the operator $U$ satisfies the following commutation relation on the low energy subspace of zero flux $da = 0, da' = 0$:

$$\text{3-group commutation relations:} \quad \begin{aligned} [V, U] &\equiv VUV^{-1}U^{-1} = W', \quad [V', U] = W, \\ [V, T] &= W', \quad [V', T] = W , \end{aligned} \tag{54}$$

where the operators are supported on the following geometry:

- In the commutator $[V, U]$, the operator $U$ is supported on the half-space $x \le 0$, and its orientation-reversal is supported on the other half space $x \ge 0$. The domain wall that separates the two half-spaces intersect the membrane that supports the operator $V'$ by a curve, and the commutation relation implies that this curve supports the Wilson line $W$. In addition, there is Wilson line $W$ on the curve at the self-intersection of the membrane that supports $V'$ in the region that supports $U$. Similarly, there is Wilson line $W'$ on the curve at the self-intersection of the membrane that supports $V$ in the region that supports $U$.

- In the commutator $[V, T]$, the Wilson line $W'$ is supported at the curve on the intersection between the membranes that support $V$ and $T$. Similarly, $W$ is supported on the curve on the intersection of the membranes that support $V$ and $T'$.

The derivation of the commutation relation (54) follows from a similar derivation in the $\mathbb{Z}_2$ gauge theory example. Let us denote $\lambda$ (resp. $\lambda'$) to be a $\mathbb{Z}_2$-valued 1-cocycle that takes the nonzero value on edges intersecting the membrane operator $V$ (resp. $V'$), otherwise zero. The first commutation relation gives

$$[V, U] = (-1)^{\int (\frac{d\tilde{\lambda}}{2} + d(a \cup_1 \lambda)) \cup a'} , \tag{55}$$

which in the low energy subspace $da = 0, da' = 0$ equals

$$[V, U] = (-1)^{\int \lambda \cup \lambda \cup a'} = W'(\gamma) \ , \tag{56}$$

where $\gamma$ is the curve given by the self-intersection of the membrane in the region that supports the operator $U$.

The second commutation relation can be derived similarly:

$$[V', U] = (-1)^{\int \frac{d\tilde{a}}{2} \cup \lambda'} = (-1)^{\frac{1}{2} \int d(a \cup \lambda') + \int a \cup \lambda' \cup \lambda'} \ . \tag{57}$$

Let us first consider the case that the membranes do not self intersect, $\lambda' \cup \lambda' = 0$, then at the low energy subspace with $da = 0, da' = 0$, the integral is

$$[V', U] = (-1)^{\int_{x=0} a \cup \lambda'} = W(\gamma') \ , \tag{58}$$

where $\gamma'$ is the intersection of the membrane that supports $V'$ with the domain wall across which the orientation of $U$ is reversed.

In addition, if the membrane self intersect $\lambda' \cup \lambda' = d\tilde{\lambda}'/2 \neq 0$, there is additional Wilson line $W(\gamma'') = (-1)^{\int a \cup \frac{d\tilde{\lambda}'}{2}}$, where $\gamma''$ is at the self intersection of the membrane that supports $V'$ in the region that supports $U$.

# 4    Anomaly and correlation functions in gauge theory with bosons

In this section, we will use examples to illustrate that the symmetry generated by the magnetic defects $U_g$ of codimension-2 has a mixed anomaly with other symmetries generated by gauged SPT defect $\mathcal{W}_{\eta^{(n)}}$. In other words, the higher-group symmetry has an 't Hooft anomaly. The 't Hooft anomalies are completely described by non-trivial correlation functions of the symmetry generators that take non-trivial numerical values. We note that since the correlation functions involving the gauged SPT defect are all trivial, the symmetries generated only by the gauged SPT defects do not have an 't Hooft anomaly.

## 4.1    Example: $\mathbb{Z}_2$ gauge theory in (3+1)D

The anomaly of the 3-group symmetry discussed in the previous section is computed in Ref. [44], and the anomaly can be derived from the correlation function of the generators of the 0-form symmetry, 1-form symmetry and 2-form symmetry in the 3-group symmetry. Denote the support of the magnetic defect by $\Sigma$, which is on the boundary of $V_3$. Then the magnetic defect sources the gauge field

$$a = \delta(V_3)^{\perp} \ , \tag{59}$$

where $\delta(V_3)^{\perp}$ is the delta function 1-form that is Poincaré dual to $V_3$. The correlation function of the magnetic defect and another gauged SPT defect is given by substituting the above value of the gauge field into the topological action on the manifolds that support the gauged SPT defect. Following the notation in Section 3.1.1, let us denote the generator of the center 1-form symmetry by $V$ that is supported on a surface $\Sigma$, the generator for the 2-form symmetry by the Wilson line $W$ that is supported on a curve $\gamma$, and the generator of the 0-form symmetry that supports a gauged Levin-Gu $\mathbb{Z}_2$ SPT phase by $U$, which is supported on a three-dimensional region $R$. The correlation function of the generators is given by

$$\langle V(\Sigma) W(\gamma) U(R) \rangle = (-1)^{\int_{\gamma} \delta(V_3)^{\perp}} (-1)^{\int_R \delta(V_3)^{\perp} \cup \delta(\Sigma)^{\perp}/2} \ , \tag{60}$$

where the first sign is the braiding of $\gamma$ and $\Sigma$, and the second phase is given by the self-braiding of the intersection of the membrane $\Sigma$ on the domain wall. The correlation function can also be explained by the charge-flux attachment: the non-trivial self-braiding of the magnetic particle appears because there is non-trivial topological action on the domain wall, and the magnetic particles (which are the intersection point of the flux loop with the domain wall) carry electric charges, and thus the full braiding of the magnetic excitation on the domain wall produces a sign from the braiding of the electric charge and the flux loop.

**'t Hooft anomaly as correlation function**    The 't Hooft anomaly of 3-group symmetry $\mathbb{G}^{(3)}$ in (3+1)D can be described by an SPT phase in (4+1)D bulk with the 3-group symmetry, as classified by $H^5(B\mathbb{G}^{(3)}, U(1))$ [41]. Such bulk SPT phases can be described in terms of an effective action in the bulk that depends on the background gauge fields of the 3-group symmetry. In the following, we will express this effective action using the correlation function of the symmetry generators.

We can describe the correlation function on a coordinate patch using the background gauge fields that describe the geometries supporting the defects, using the Poincaré duality. The backgrounds are

$$C_3 = \delta(\gamma)^\perp, \quad B_2 = \delta(\Sigma)^\perp = d\lambda, \quad \lambda \equiv \delta(V_3)^\perp, \quad C_1 = \delta(R)^\perp . \tag{61}$$

The correlation function in terms of the backgrounds is:

$$\exp\left( \pi i \int \lambda \cup C_3 + \frac{\pi i}{2} \int \lambda \cup d\lambda \cup C_1 \right) = \exp\left( \pi i \int_{\text{bulk}} \left( d\lambda \cup C_3 + \lambda \cup dC_3 + \frac{1}{2} d\lambda \cup d\lambda \cup C_1 + \frac{1}{2} \lambda \cup d\lambda \cup dC_1 \right) \right) , \tag{62}$$

where we rewrite the integral on the left-hand side in terms of a $(D+1)$-dimensional bulk manifold that bounds the spacetime. Using the 3-group relation for the background fields $dC_3 = B_2 \cup \text{Bock}(C_1) = B_2 \cup d\tilde{C}_1/2$, we find the correlation function (60) is given by $e^{iS_{\text{bulk}}}$ with

$$S_{\text{bulk}} = \pi \int_{\text{bulk}} B_2 \cup C_3 + \frac{\pi}{2} \int_{\text{bulk}} \mathcal{P}(B_2) \cup C_1 , \tag{63}$$

where $\mathcal{P}(B_2) = \tilde{B}_2 \cup \tilde{B}_2 - B_2 \cup_1 dB_2$ is the Pontryagin square operation that takes $\mathbb{Z}_2$ valued 2-cocycle and maps it to a $\mathbb{Z}_4$ valued 4-cocycle. On a coordinate patch, $B_2 = d\lambda$, and the bulk term can be written as a boundary term. The complete bulk action is given by evaluating the action and gluing $B_2 = d\lambda$ with a suitable transition function across the coordinate patches to obtain non-trivial $B_2$ by the analogue of the clutching construction for vector bundles. The bulk action describes an SPT phase with 3-group symmetry $\mathbb{G}^{(3)}$, which is classified by $H^5(B\mathbb{G}^{(3)}, U(1))$.

As discussed above, the correlation function (60) of the magnetic defect $V$, the Wilson line $W$ and the domain wall $U$ is given by $e^{iS_{\text{bulk}}}$. In particular, the first term in (63), $\int \delta(V_3)^\perp \delta(\gamma)^\perp = \text{Link}(\Sigma, \gamma)$ with Link for the linking number, represents the mutual braiding between the electric particle and the magnetic flux loop. The second term represents the framing anomaly of the magnetic particle on the codimension-1 defect due to its semion self-statistics.

### 4.1.1 Lattice description of the anomaly in $\mathbb{Z}_2$ gauge theory

The anomaly can be described by the following commutation relations

$$[V, W] = -1, \quad [V, [V, U]] = -1 . \tag{64}$$

The first equation is valid in the whole Hilbert space of the lattice model. The second equation is valid only within the low energy subspace with the zero flux, and thus should be regarded as an anomaly of emergent symmetries on the low energy subspace. This reflects that while the operators $V, W$ commute with the Hamiltonian and generate global symmetries in the whole Hilbert space, $U$ only works as an emergent symmetry of low-energy subspace with zero flux.

The first commutator represents the braiding of the Wilson line and the magnetic defect, with the membrane that supports $V$ having an odd number of intersections with the Wilson line $W$. The second commutator follows from $[V, U] = W$, and the first commutator, and it represents the braiding between the extra electric charge created by $W$ carried by the magnetic excitation, and the magnetic excitation (where the magnetic excitations are separated also in time due to the ordering in the commutation relation). In other words, the full mutual braiding between magnetic excitations on the domain wall produces a sign.

The anomaly precludes the presence of a symmetry-preserving gapped ground state, which is consistent with the fact that the ground state subspace spontaneously breaks the emergent 3-group symmetry on topologically non-trivial manifolds.

## 4.2 Example: untwisted $\mathbb{Z}_2 \times \mathbb{Z}_2$ gauge theory in (3+1)D

We can repeat the discussion for $\mathbb{Z}_2 \times \mathbb{Z}_2$ gauge theory. For simplicity, let us focus on the "subgroup" 3-group symmetry that only involves the center 1-form symmetry, generated by the magnetic defects $V, V'$, the 2-form symmetry generated by the Wilson lines $W, W'$, and the 0-form symmetry generated by the domain wall $U$ that hosts a gauged SPT phase with $\mathbb{Z}_2 \times \mathbb{Z}_2$ symmetry: in terms of the $\mathbb{Z}_2 \times \mathbb{Z}_2$ gauge fields $a, a'$, the symmetry generators are

$$W(\gamma) = (-1)^{\oint_\gamma a}, \quad W'(\gamma') = (-1)^{\oint_{\gamma'} a'}, \quad U(R) = (-1)^{\oint_R a \cup d\tilde{a}'/2} , \tag{65}$$

in addition to the magnetic defects $V, V'$ supported on surfaces $\Sigma, \Sigma'$, which are on the boundary of $V_3, V_3'$, respectively. The magnetic defects source the gauge field

$$a = \delta(V_3)^\perp, \quad a' = \delta(V_3')^\perp . \tag{66}$$

The correlation function on a coordinate patch can be computed by substituting (66) in the topological actions on the gauged SPT defects: using (65), we find

$$\langle U(R)V(\Sigma)V'(\Sigma')W(\gamma)W'(\gamma')\rangle = (-1)^{\int_\gamma \delta(V_3)^\perp}(-1)^{\int_{\gamma'} \delta(V_3')^\perp}(-1)^{\int_R \delta(V_3)^\perp \cup \delta(\Sigma')^\perp/2} . \tag{67}$$

The first two terms describe the braiding of the magnetic defects with the Wilson lines, while the last term describes the mutual braiding of the magnetic defects $V, V'$ on the domain wall $R$, where the magnetic defects intersect the domain wall by 1-dimensional loops.

**Anomaly from correlation function**   In terms of the background fields,

$$C_3 = \delta(\gamma)^\perp, \quad C_3' = \delta(\gamma')^\perp, \quad B_2 = d\lambda, \quad B_2' = d\lambda', \quad \lambda \equiv \delta(V_3)^\perp, \quad \lambda' = \delta(V_3')^\perp, \quad C_1 = \delta(R)^\perp , \tag{68}$$

the correlation function is $e^{iS_{\text{bulk}}}$ with

$$S_{\text{bulk}} = \pi \int_{\text{bulk}} \left( (d\lambda \cup C_3 + d\lambda' \cup C_3' + \lambda \cup dC_3 + \lambda' \cup dC_3') + \frac{1}{2} d\lambda \cup d\lambda' \cup C_1 + \frac{1}{2}\lambda \cup d\lambda' \cup dC_1 \right)$$

$$= \pi \int_{\text{bulk}} (B_2 \cup C_3 + B_2' \cup C_3') + \frac{\pi}{2} \int_{\text{bulk}} (\tilde{B}_2 \cup \tilde{B}_2' - dB_2 \cup_1 B_2') \cup C_1 . \tag{69}$$

where we used $dC_3 = B_2' \cup \frac{d\tilde{C}_1}{2} + \frac{d\tilde{B}_2'}{2} \cup C_1$, $dC_3' = \frac{d\tilde{B}_2}{2} \cup C_1$, with tilde denotes a lift of the gauge field to $\mathbb{Z}_4$ value.

As discussed above, the correlation function (67) of the magnetic defects $V, V'$, the Wilson lines $W, W'$, and the domain wall $U$ is given by $e^{iS_{\text{bulk}}}$. In particular, the first term in $S_{\text{bulk}}$, $\int \delta(V_3)^\perp \delta(\gamma)^\perp = \text{Link}(\Sigma, \gamma)$ and $\int \delta(V_3')^\perp \delta(\gamma')^\perp = \text{Link}(M_2', \gamma')$, represents the mutual braiding of electric particles and magnetic flux loops for the two $\mathbb{Z}_2$ gauge fields. The second term in $S_{\text{bulk}}$ describes the braiding between magnetic particles (which acquire fractional statistics) induced on the domain wall due to intersection of the domain wall with magnetic flux loops.

# 5   Application: higher-group symmetry and Clifford hierarchy of logical gates

In this section, we will explain that the higher-group symmetry of Hamiltonian models gives non-trivial constraints on the logical gates realized by the symmetry generators on the ground state subspace.

We will illustrate this with the stabilizer Hamiltonian, where the Hilbert space is given by $N$ types of qubits on each edge of the lattice. The $\mathbb{Z}_2^N$ gauge theory is realized as the low energy subspace of the topological stabilizer model, where the ground state subspace is referred to as the codespace and encodes logical qubits. A unitary operator acting within the codespace is referred to as a logical gate. The symmetry generators preserve the low energy subspace, and they are logical gates on the codespace.

We will show that higher-group symmetry commutation relation gives rise to new logical gates in $\mathbb{Z}_2^N$ toric code model, such as control-$Z$ (CZ) gate in (3+1)D $\mathbb{Z}_2$ toric code. In Ref. [93], it is stated that the symmetry generators of $(d + 1)$-dimensional $\mathbb{Z}_2^N$ gauge theory is bounded by $d$-th level of the Clifford hierarchy denoted as $\mathcal{P}_d$. We will show that the such bound can in fact be improved by considering the gauged SPT defects and their higher-group commutation relations.

## 5.1   Control-Z gate in (3+1)D $\mathbb{Z}_2$ toric code from Klein bottle

In the lattice Hamiltonian models for the $\mathbb{Z}_2$ gauge theory in (3+1)D discussed in Section 3, we have seen that the 3-group symmetry leads to the following commutation relation between symmetry generators of the form

$$[U, V(\Sigma)] = W(\gamma), \tag{70}$$

together with $[U, W(\gamma)] = 1$, where $V(\Sigma)$ represent the magnetic defects that generate the $\mathbb{Z}_2$ 1-form symmetry, and $W(\gamma)$ is the electric Wilson line operator generating the $\mathbb{Z}_2$ 2-form symmetry. Note that the above commutator gives a non-trivial logical gate if and only if $U$ supports an orientation-reversing defect on it, and $\Sigma$ intersects the orientation-reversing defect along a topologically non-trivial 1-cycle $\gamma$.

This can happen when we have a non-orientable 3d space such as Klein bottle $\times S^1$. Under these conditions, the magnetic defect $V(\Sigma)$ is then identified as the Pauli $X$ gate acting on the codespace, while $W(\gamma)$ is regarded as the Pauli $Z$ gate. The above commutation relation then implies that the SPT defect $U$ encodes CZ logical gate acting on the codespace. There are 3 logical qubits, and depending on where we put the orientation reversal the CZ gate acts on two of them.

## 5.2 Review of Clifford hierarchy

More generally, in lattice models that realize $\mathbb{Z}_2^N$ gauge theories for general $N$, the sets $\{\mathcal{P}_j\}$ of logical gates labeled by positive integer $j$ that describe the symmetry generators are organized inductively by a structure called Clifford hierarchy [93, 133]:

- $\mathcal{P}_1$ is the group generated by all Pauli $X, Z$ gates acting on the codespace

- $\mathcal{P}_j$ for higher $j$ is defined as the set of unitary operators $U$ satisfying $U\mathcal{P}_1 U^\dagger \subseteq \mathcal{P}_{j-1}$ for $j \geq 2$

The set $\mathcal{P}_j$ forms a group for $j = 1, 2$, where $\mathcal{P}_1$ is called Pauli group and $\mathcal{P}_2$ is called Clifford group. $\mathcal{P}_j$ for $j \geq 3$ does not have a group structure and constitutes the $j$-th level of Clifford hierarchy.

## 5.3 Relation between Clifford hierarchy and higher-group symmetry in (3+1)D

Let us discuss the relation between 3-group symmetry in $\mathbb{Z}_2^N$ gauge theories in (3+1)D, and the Clifford hierarchy in the corresponding lattice Hamiltonian model.

First, we will show that the action of 1-form symmetry of (3+1)D $\mathbb{Z}_2^N$ gauge theory given by (1+1)D SPT phase is contained in a certain subgroup $\mathcal{P}_2'$ of the Clifford group $\mathcal{P}_2$, where $\mathcal{P}_2'$ is generated by $S, CZ$ gates and overall $U(1)$ phase $e^{i\phi} \in U(1)$ acting on the codespace.

This can be established by computing the commutation relation between the 1-form symmetry generator $U_1(\Sigma)$ and logical $X, Z$ operators realized by $V(\Sigma), W(\gamma)$ with $\gamma, \Sigma$ the non-trivial 1, 2-cycles of the manifolds respectively. The symmetry generator $U_1(\Sigma)$ given by SPT phase commutes with $W(\gamma)$, so $[U_1(\Sigma), Z] = 1$ for all Pauli $Z$ operators. Also, the commutation relation between $U_1(\Sigma)$ and $V(\Sigma')$ is in general given by the 2-form symmetry generator supported on the intersection of two surfaces $\Sigma$ and $\Sigma'$, so it implies that $[U_1(\Sigma), X]$ has the form of

$$[U_1(\Sigma), X] = \pm Z \text{ or } \pm 1, \tag{71}$$

since a non-trivial generator of 2-form symmetry is generally given by the electric line operator $W(\gamma)$ that corresponds to a $Z$ gate. The above commutation relations involving $X, Z$ are satisfied if and only if $U_1(\Sigma)$ is the element of $\mathcal{P}_2'$.

In particular, the non-trivial commutation relation $[U_1(\Sigma), X] = \pm Z$ implies that the 1-form and 2-form symmetry forms a 3-group. Therefore, in order for $U(\Sigma)$ to realize the non-Pauli Clifford gate in $\mathcal{P}_2' \setminus \mathcal{P}_1$, it is necessary that the 1-form symmetry $U_1, V$ and 2-form symmetry $W$ mix into 3-group.

Let us then look at the generator $U_0$ of the 0-form symmetry given by (2+1)D SPT phase, which fills the entire 3d space. The commutation relations involving $Z, X$ are in general given in the form of

$$[U_0, X] \subseteq \mathcal{P}_2', \quad [U_0, Z] = 1 \tag{72}$$

The set of unitary operators satisfying the above commutation relation defines a subset $\mathcal{P}_3'$ of $\mathcal{P}_3$. The possible action of 1-form symmetry in $\mathbb{Z}_2^N$ gauge theory is bounded by the set $\mathcal{P}_3'$. When the commutation relation is given in the form of $[U_0, X] = \pm Z$, it implies that the 0-form, 1-form and 2-form symmetries together form a 3-group, as we have seen in previous sections. Meanwhile, when $[U_0, X]$ is given by a non-trivial Clifford gate in $\mathcal{P}_2' \setminus \mathcal{P}_1$, it implies that the 0-form symmetry induces the permutation of 1-form symmetry defects. In that case, the 0-form symmetry $U_0$ encodes the non-Clifford gate in $\mathcal{P}_3' \setminus \mathcal{P}_2'$. [14]

## 5.4 Constraints on logical gates in general dimensions from higher-group symmetry

### 5.4.1 Higher-group symmetry in untwisted $\mathbb{Z}_2^N$ gauge theory on the lattice in general dimension

Let us consider the lattice Hamiltonian model for untwisted $G = \mathbb{Z}_2^N$ gauge theory on $d$-dimensional hypercubic Euclidean lattice $\mathbb{Z}^d$. On each edge we place $N$ qubits, acted on by the Pauli matrices $X^{(I)}, Y^{(I)}, Z^{(I)}$ with $I = 1, \cdots, N$. We can obtain a non-trivial logical code subspace by picking appropriate boundary conditions, although the precise details are not important for our discussion below.

The theory has higher-group symmetry that mixes the symmetries generated by the gauged SPT defects and the 1-form symmetry generated by the magnetic defects, and such higher-group symmetry leads to the non-trivial algebra on the logical gates described by the symmetry generators on the ground state subspace. The symmetry generators are

---

[14] An example of the logical gate in $\mathcal{P}_3' \setminus \mathcal{P}_2'$ is found in (3+1)D $\mathbb{Z}_2^3$ toric code, where the 0-form symmetry is generated by the type-III cocycle in $H^3(B\mathbb{Z}_2^3, U(1))$ [90].

- 1-form symmetry generated by the magnetic defect, represented by the operator that is the product of Pauli $X$ gates. For the magnetic defect that carries holonomy $g = (g_1, g_2, \cdots, g_N) \in \mathbb{Z}_2^N$,

$$V_g(\Sigma) = \prod_{I=1}^{N} \prod_e (X_e^{(I)})^{g_I} , \qquad (73)$$

where the product is over the edges that cut the codimension-one subspace $\Sigma$ on the dual lattice. The superscript $I$ labels the Pauli operators that act on different qubits, and it is included in the product if and only if the $I$th component of $g \in \mathbb{Z}_2^N$ is the non-trivial $\mathbb{Z}_2$ flux.

- $n$-form symmetry generated by gauged SPT defect. For the symmetry group element $[\eta^{(d-n)}] \in H^{d-n}(BG, U(1))$, the corresponding operator is

$$U_\eta(R, \sigma) = e^{i \int a^* \eta^{(d-n)}} , \qquad (74)$$

where $a = (a^{(1)}, a^{(2)}, \cdots a^{(N)})$ is the gauge field for $G = \mathbb{Z}_2^N$, with $a^{(I)}(e) = (1 - Z_e^{(I)})/2$, and $\sigma$ assigns orientation to all simplices in the $(d-n)$-dimensional subspace $R$ on the lattice. The operator can be described in the $Z^{(I)}$-eigenbasis for all $I = 1, \cdots, N$. The integral $\int a^* \eta^{(d-n)}$ is given by summing over the contributions from $(d-n)$-dimensional simplices, where on each $(d-n)$-simplex it is given by the value of $a^* \eta^{(d-n)}$ for the field specified by the eigenvalue of $a(e)$ on each edge $e$ in the simplex.

These symmetries mix into a higher group, which can be shown from the commutation relation (where we omit an overall phase):

$$[U_\eta(R, \sigma), V_g(\Sigma)] = U_{i_g^A \eta}(R_A, \sigma) \cdot U_{i_g^B \eta}(R_B, \sigma) , \qquad (75)$$

where $U_\eta(R, \sigma)$ generates an $n$-form symmetry, $U_{i_g^A \eta}(R_A, \sigma)$ generates an $(n+1)$-form symmetry, and $U_{i_g^B \eta}(R_B, \sigma)$ generates an $(n+2)$-form symmetry. We take $\Sigma$ to be a codimension-1 hyperplane without self-intersection on the dual lattice. On the right-hand side of the commutation relation,

- $U_{i_g^A \eta}(R_A, \sigma)$is supported on region $R_A$, where $R_A$ is the intersection between $\Sigma$ and $R$. This contribution is discussed in Ref. [90]. Examples of such commutation relation are the commutator $[V, T] = W'$ and $[V', T] = W$ in Eq. (54) in (3+1)D $\mathbb{Z}_2^2$ toric code model in Section 3.1.

  The commutation relation implies that the intersection of the symmetry generators $V, U_\eta$ for the 1-form and $n$-form symmetries produces the generator $U_{i_g^A \eta}$ of $(n+1)$-form symmetry. Such higher-group symmetry can be described in terms of the background gauge fields $B_2, C_{n+1}, C_{n+2}$ of the symmetries by the relation of the form $dC_{n+2} \supset B_2 \cup C_{n+1}$, such as the first term on the right-hand side of the last two lines in Eq. (48) in (3+1)D $\mathbb{Z}_2^2$ toric code model in Section 3.1.

- The $(d-n)$-dimensional subspace $R$ has a $(d-n-1)$-dimensional orientation-reversing defect on it, across which the orientation $\sigma$ changes. $U_{i_g^B \eta}(R_B, \sigma)$ is supported on the region $R_B$ given by the intersection between $\Sigma$ and the orientation-reversing defect in $R$. Examples of such commutation relation are Eq. (42) and the first line in Eq. (54) in the (3+1)D $\mathbb{Z}_2$ and $\mathbb{Z}_2^2$ toric code models in Section 3.1.

  The commutation relation implies that the intersection of the symmetry generators $V, U_\eta$ for the 1-form and $n$-form symmetries produces the generator $U_{i_g^B \eta}$ of $(n+2)$-form symmetry. Such higher-group symmetry can be described by the relation for background gauge fields of the form $dC_{n+3} \supset B_2 \cup \text{Bock}(C_{n+1})$, such as Eq. (35) and Eq. (48) in the (3+1)D untwisted $\mathbb{Z}_2$ and $\mathbb{Z}_2^2$ gauge theories in Section 3.1.

### 5.4.2 Consequences for Clifford hierarchy and comparison with the literature

The above commutation relation implies that the $(d-n)$-dimensional gauged SPT defect $U_\eta$ gives an element of $\mathcal{P}'_{d-n}$, where $\mathcal{P}'_k$ for $k \geq 3$ is defined inductively as the set of unitary operators $U$ satisfying $[U, X] \subseteq \mathcal{P}'_{k-1}$ and $[U, Z] = 1$, with $\mathcal{P}'_2$ defined earlier by the subset generated by the $S, CZ$ gates and overall $U(1)$ phase $e^{i\phi} \in U(1)$. This bound on the possible action of SPT logical gates $U_\eta \subseteq \mathcal{P}'_{d-n}$ for $0 \leq n \leq d-1$ refines the bound given in Ref. [93], where it states that symmetry defect of $(d+1)$-dimensional $\mathbb{Z}_2^N$ gauge theory is bounded by $\mathcal{P}_d$.

# 6  $d$-group symmetry in $(d+1)$D Dijkgraaf-Witten gauge theory with bosonic charges

## 6.1  $d$-group symmetry from 1-form transformations and generalized Witten effect

In the general bosonic gauge theory with boson particles, let us consider the symmetry generated by the defect with gauged SPT phase supported on $n$-dimensional submanifold, and the symmetry generated by the magnetic defect:

$$\mathcal{G}^{(n)} = H^{D-n-1}(BG, U(1)), \text{ for } n \neq 0, 1$$
$$\mathcal{G}^{(0)} = H^{D-1}(BG, U(1)) \rtimes S,$$
$$\mathcal{G}^{(1)} = \text{extension of } Z_\omega(G) \text{ by } H^{D-2}(BG, U(1)) , \tag{76}$$

where the extension is defined by a 2-cocycle $[\Omega] \in H^2(BZ_\omega(G), H^{D-2}(BG, U(1)))$, given by

$$\Omega(g, g') = \left( i_g^B \omega^{(D)} + i_{g'}^B \omega^{(D)} - i_{g+g'}^B \omega^{(D)} \right) / |\omega^{(D)}|. \tag{77}$$

$S \subset \text{Aut}(G)$ leaves the topological action $\omega^{(D)}$ invariant.

In general, the 0-form symmetry $\mathcal{G}^{(0)}$ can permute the generators of other symmetries $\mathcal{G}^{(k)}$ using the automorphisms in $S$, and also dress the magnetic defects in $\mathcal{G}^{(1)}$ with gauged SPT defects. We thus have a set of group homomorphisms:

$$\rho_n : \mathcal{G}^{(0)} \to \text{Aut}(\mathcal{G}^{(n)}), \ \ n \geq 1, \tag{78}$$

such that for $n > 1$, the permutation action arises entirely from the automorphisms in $S$. For $n = 1$, the group homomorphism describes how a codimension-2 magnetic defect is permuted into a combination of a magnetic defect and a codimension-2 gauged SPT defect as it crosses a codimension-1 gauged SPT domain wall, as can be determined by dimensional reduction and the slant product. In the discussion below we will omit $S$ for simplicity.

When the magnetic defect intersects the gauged SPT defect, it creates a magnetic source on the submanifold of some dimension $n$ that supports the gauged SPT defect. Due to the charge-flux attachment or generalized Witten effect, this magnetic source is attached to a new gauged SPT defect of dimension $(n-1)$. In other words, a new gauged SPT defect of lower dimension is emitted from such a junction. The property that the intersection of the magnetic defect and the original gauged SPT defect that generate 1-form and $(D-n-1)$-form symmetry creates another gauged SPT defect of lower dimension that generates higher-form symmetry is the hallmark of higher-group symmetry (see *e.g.* [40, 41]): the higher-form symmetry mixes with the lower form symmetries. If we wish to study the fractionalization [23, 134, 41, 57] of only the lower form symmetry part of the higher group, this represents an obstruction to symmetry localization: we cannot only consider defects of the lower form symmetries, since some configuration of defect intersection necessarily produces the defects that generate the higher-form symmetry. All of these follow from the generalized Witten effect or the charge-flux attachment. Thus higher-group symmetry is very generic in gauge theory in (3+1)D and higher spacetime dimensions. It also applies to higher-form gauge theories, which also have the generalized Witten effect from fiber integration.[15]

Let us phrase the above description using background gauge fields of the symmetries. The emission of the gauged SPT defect at various junctions in the presence of the magnetic defect indicates that performing the corresponding 1-form transformation for the 1-form symmetry generated by the magnetic defect produces an extra background gauge field for the symmetry generated by the gauged SPT defects, and such mixing of transformation is the hallmark of a higher-group symmetry [40, 41]. Let us turn on the background gauge fields $C_n$ for the symmetries generated by the gauged SPT defects. Denoting the dynamical $G$ gauge field by $a$, the action is

$$S[a, \{C_n\}] = \int \left( C_n \cup a^* \eta^{(D-n)} + a^* \omega^{(D)} \right), \ \ \eta^{(n)} \in H^n(BG, U(1)), \ \ \omega^{(D)} \in H^D(BG, U(1)) , \tag{79}$$

where $a^* \eta^{(n)}$ is the pullback of the group cocycle $\eta^{(n)}$ by the gauge field $a$. For the $(n-1)$-form symmetry $H^{D-n}(BG, U(1)) = \prod_i \mathbb{Z}_{N_i}$ with basis $\eta^{(D-n)} = (\eta^{(D-n)})_i$ that generates the $\mathbb{Z}_{N_i}$ subgroup, we turn on background gauge field $C_n = (C_n)^i$ where $(C_n)^i$ is the background gauge field for the $\mathbb{Z}_{N_i}$ subgroup symmetry generated

---

[15]In $n$-form gauge theory with topological action, the magnetic defect is attached to gauged SPT defect by considering $S^n$ fibration over $[0, \infty)$ with the magnetic defect placed at the origin surrounded by $S^n$ (that carry holonomy of the $n$-form gauge field) and extending in the remaining spacetime directions. Then performing fiber integration of the topological action produces the gauged SPT defect attached to the magnetic defect.

by $(\eta^{(D-n)})_i$. For simplicity, we will take the spacetime to be a $D$-dimensional torus. The higher-group symmetry involves the 1-form symmetry generated by the magnetic defect, which is a shift symmetry of the gauge field. Consider

$$a \rightarrow a + \lambda \,, \tag{80}$$

where $\lambda$ is a $Z_\omega(G)$-valued 1-form, and let us expand it as

$$\lambda = \sum_{g \in Z_\omega(G)} g\lambda^g \,, \tag{81}$$

for some basis integer 1-forms $\lambda^g$ that takes value $0, 1$ on basis 1-cycles. Let us take $d\lambda^g = 0$. In fact, on spacetime that is a $D$-dimensional torus, we can take $d\lambda^g = 0$ in real coefficient. In such case, there is no background for the center 1-form symmetry, $B_2 = 0$. We note that $B_2 = gd\lambda^g$ is equivalent to inserting a magnetic defect that carries holonomy $g$ at $M_{D-2} = \partial V_{D-1}$ with $\lambda^g = \delta(V_{D-1})^\perp$. Thus for $B_2 = 0$, we take $V_{D-1}$ to have no boundary. In other words, we first insert magnetic defect on the boundary of some open submanifold $V'_{D-1}$, then remove the magnetic defect by closing up the boundary of $V'_{D-1}$ to change $V'_{D-1}$ into a closed submanifold $V_{D-1}$. For the action to be invariant up to terms that depend on the classical fields $\lambda$ and $C_k$ but independent of the dynamical field $a$ (such terms describe the 't Hooft anomaly), the backgrounds in general must transform in a non-linear way. To see this, we note that the cocycle changes by

$$(a+\lambda)^*\eta^{(n)} - a^*\eta^{(n)} = \lambda^*\eta^{(n)} + \sum_g \lambda^g \cup a^*(i_g\eta^{(n)}) + \sum_{g_1,g_2} \lambda^{g_1} \cup \lambda^{g_2} \cup a^*(i_{g_2}i_{g_1}\eta^{(n)})$$
$$+ \cdots \sum_{\{g_i\}} \lambda^{g_1} \cup \lambda^{g_2} \cup \cdots \lambda^{g_{n-1}} \cup a^*(i_{g_{n-1}}i_{g_{n-2}}\cdots i_{g_1}\eta^{(n)}) \,. \tag{82}$$

We can verify the relation by integrating the cocycle over arbitrary $n$-dimensional torus, where $\lambda$ changes the holonomy of the gauge field (which is flat for finite gauge group) on the circles inside the torus. Concretely, for each 1-cycle $\gamma_j$ of the torus, one can rewrite the integral on a torus with holonomy $a_j + \lambda_j$ in the direction of $\gamma_j$ as the integral on the disconnected sum of two tori with holonomy $a_j$ and $\lambda_j$ respectively. By performing this process for each 1-cycle of the torus, one can decompose $\int_{T^n} (a+\lambda)^*\eta^{(n)}$ into the right-hand side of (82) in addition to $\int_{T^n} a^*\eta^{(n)}$, which verifies (82).

Thus the term $C_n \cup a^*\eta^{(n)}$ produces terms with lower degree group cocycles, and to compensate for the change, the backgrounds of higher degree that couple to these lower cocycles must be shifted. Such mixing of transformation is the hallmark of higher-group symmetry, or the symmetry extension involving symmetry of different form degrees. The correction can be obtained by iteration, starting from correcting the transformation of $C_2, C_3, \cdots, C_{D-1}$ by non-trivial shift. To compute the shift, we also recall that $i_g\eta^{(n)}$ can be decomposed into two parts

$$i_g\eta^{(n)} = i_g^A\eta^{(n)} + \frac{1}{|\eta^{(n)}|}d(i_g^B\eta^{(n)}) \,, \tag{83}$$

where $|\eta^{(n)}|$ is the order of the class $[\eta^{(n)}]$ in $H^n(BG, U(1))$. We will see that the second term, $i_g^B\eta^{(n)}$, does not contribute to the shift of the background gauge fields $C_k$ when the transformation parameter satisfies $d\lambda^g = 0$ in real coefficients.

Let us denote the shifted backgrounds by $C_n^\lambda$, which differ from $C_n$ by a shift induced from the $\lambda$ transformation. We note that $C_1$ does not need to be shifted, $C_1^\lambda = C_1$.

The first case of correction on the transformation starts from $C_2$ that couples to the gauged SPT defect of dimension $(D-2)$. The background $C_2$ needs to be shifted as

$$C_2 \rightarrow C_2^\lambda = C_2 - C_1 \cup \sum_g \lambda^g M^{(D-2)}(i_g^A\eta^{(D-1)}) \,, \tag{84}$$

where we expand the backgrounds $C_n$ under some basis in $H^{D-n+1}(BG, U(1))$, and $M^{(n)}(x)$ for $x \in H^n(BG, U(1))$ is the expansion of $x$ under the basis of $H^n(BG, U(1))$.[16]

The physical meaning of the above transformation of $C_2$ is the following. The fact that the transformation shifts the background $C_2$ means that different ways of adding the magnetic defect can produce a gauged SPT defect that

<hr/>

[16] We note that if we consider $g \in Z(G)$ but not in $Z_\omega(G)$, then there will be extra term $i_g^A i_{g'}^A \omega^{(D)}$ that arises from the property that the magnetic defect is attached to a gauged SPT defect as in Figure 1, which vanishes for $g, g' \in Z_\omega(G)$. Also, if $d\lambda^g/|\omega^{(D)}| \neq 0$, there is extra term $-\sum_g \frac{1}{|\omega^{(D)}|}d\lambda^g M^{(D-2)}(i_g^B\omega^{(D)})$ that arises from the property that the junction of the magnetic defect emits a gauged SPT defect as in Figure 2.

couples to the background $C_2$. From the first term with $C_1$, if we intersect the magnetic excitation with the domain wall for $C_1$, then there is additional excitation created by the gauged SPT defect of dimension $(D-2)$. This is the statement about the 0-form symmetry acts on the 1-form symmetry: when a magnetic defect of holonomy $g$ crosses the domain wall, it is stacked with additional gauged SPT defect of dimension $(D-2)$.

Let us continue to the correction for the transformation of $C_3$:

$$C_3 \to C_3^\lambda = C_3 - C_2^\lambda \cup \sum_g \lambda^g \cup M^{(D-3)}(i_g^A \eta^{(D-2)}) - C_1 \cup \sum_{g_1,g_2} \lambda^{g_1} \cup \lambda^{g_2} M^{(D-3)}(i_{g_2}^A i_{g_1}^A \eta^{(D-1)})$$
$$- \sum_{g_1,g_2,g_3} \lambda^{g_1} \cup \lambda^{g_2} \cup \lambda^{g_3} M^{(D-3)}(i_{g_3}^A i_{g_2}^A i_{g_1}^A \omega^{(D)}) \ . \tag{85}$$

Continuing the induction process, if we already computed the shifted backgrounds $C_2^\lambda, C_3^\lambda, \cdots, C_{n-1}^\lambda$, then the correction to $C_n$ is

$$C_n \to C_n^\lambda = C_n - C_{n-1}^\lambda \cup \sum_g \lambda^g M^{(D-n)}(i_g^A \eta^{(D-n+1)}) - C_{n-2}^\lambda \cup \sum_{g_1,g_2} \lambda^{g_1} \cup \lambda^{g_2} M^{(D-n)}(i_{g_2}^A i_{g_1}^A \eta^{(D-n+2)}) - \cdots$$
$$\cdots - \sum_{g_1,\cdots g_n} \lambda^{g_1} \cup \cdots \lambda^{g_n} M^{(D-n)}(i_{g_n}^A i_{g_{n-1}}^A \cdots i_{g_1}^A \omega^{(D)}) \ . \tag{86}$$

Such mixture of transformation for background gauge fields describe a $(D-1)$-group symmetry (if $C_{D-1}^\lambda \neq C_{D-1}$ is corrected). The physical meaning of each term in the shift of $C_n$ is different way of producing an extra gauged SPT defect that couples to $C_n$ by adding a magnetic defect.

## 6.2   Description of Postnikov classes

We can describe the modification of the background gauge transformations as modified cocycle condition

$$dC_n = f_{n+1}(B_2, \{C_k\}_{k=1}^{n-1}) \ , \tag{87}$$

where the right hand side is the Postnikov class of the higher group. Denote the background for the 1-form symmetry generated by the magnetic defects by $B_2$. Under the 1-form transformation $B_2 \to B_2 + d\lambda$, $a \to a + \lambda$, the background gauge field $C_n$ is shifted by the anomaly descendant $\alpha_n$ of $f_{n+1}$, i.e. $d\alpha_n = f_{n+1}|_{B_2+d\lambda} - f_{n+1}|_{B_2}$.[17] Then $\alpha_n$ with $d\lambda = 0$ is the shift transformation of the background $C_n$ in (86). From a similar discussion as in Section 2, $\alpha_n|_{d\lambda=0}$ is obtained from the reduction of $f_{n+1}$ on a circle with $B_2 = \lambda d\vartheta/2\pi$, where $\vartheta \sim \vartheta + 2\pi$ is the angular coordinate on the compactified circle.

To obtain $f_{n+1}$ from $\alpha_n$, let us start with $B_2 = 0$, then since $f_{n+1}|_{B_2=0} = 0$, we have $f_{n+1}|_{B_n=d\lambda} = d\alpha_n$. This is the expression of $f_{n+1}$ on coordinate patches, where $B_2 = d\lambda$. Then we can obtain $f_{n+1}$ for general $B_2$ by the analogue of the clutching construction of vector bundles.[18]

Alternatively, we can infer the Postnikov classes from the junction of symmetry defects using the charge-flux attachment: when a magnetic defect intersects a gauged SPT defect, there is a lower-dimensional gauged SPT defect emitted at the intersection as specified by the slant product. The Postnikov class is an element of $[f_{n+1}] \in H^{n+1}(B\mathbb{G}^{(n-1)}, \mathcal{G}^{(n-1)})$. For $(n-1)$-group $\mathbb{G}^{(n-1)}$, which in general involves $k$-form symmetries for $0 \le k \le n-2$, each element in $H^{n+1}(B\mathbb{G}^{(n-1)}, \mathcal{G}^{(n-1)})$ is the symmetry defect of $\mathcal{G}^{(n-1)}$ that emits from a codimension-$(n+1)$ junction of these $k$-form symmetry defects (where $0 \le k \le n-2$).

The Postnikov class for the $d$-group symmetry in Dijkgraaf-Witten theory is the element of $H^{n+1}(B\mathbb{G}^{(n-1)}, \mathcal{G}^{(n-1)})$, such that the emission of the symmetry defect at the junction is specified by the charge-flux attachment. By scanning the possible defect configurations that can produce the gauged SPT defects for the $(n-1)$-form symmetry, inductively starting from $n = 1$, we obtain the Postnikov classes $f_{n+1}$ for the $d$-group symmetry.

---

[17]We note that $f_{n+1}$ can be regarded as the effective action of an SPT phase in $(n+1)$ dimension, and under the 1-form transformation by $\lambda$, $C_n \to C_n + \alpha_n$ is the anomalous transformation on the boundary.

[18]For a general transformation by $\lambda$ that might not be closed,

$$\alpha_n = (\alpha_n)|_{d\lambda=0} + d\lambda \cup \beta_{n-2} \ . \tag{88}$$

The last term $\beta$ contributes to $f_{n+1}$ by a total derivative, using $dB_2 = 0$:

$$B_2 \cup d\beta_{n-2} = d(B_2 \cup \beta_{n-2}) \ , \tag{89}$$

which can be absorbed into the definition of $C_n$, and such redefinition has the effect of changing the fractionalization class [23, 134, 41, 57]. This gives the $f_{n+1}$ from $(\alpha_n)|_{d\lambda=0}$ in (86) up to redefinitions of the background fields. As such, we set $\beta_{n-2} = 0$ in our computation.

For instance, $f_{n+1}$ for several lower values of $n$ are described as follows. For $n = 1$, there is no shift in the background $C_1$, $C_1^\lambda = C_1$. Similarly, there is no configuration of symmetry defect that can emit a domain wall decorated with gauged SPT phases. Thus

$$dC_1 = f_2 = 0 \ . \tag{90}$$

### 6.2.1   $n = 2$, extension of 1-form symmetry

We have

$$f_3 = -d(i^B_{B_2}\omega^{(D)})/|\omega^{(D)}| - i^A_{B_2}(C_1) \ , \tag{91}$$

where on a 3-simplex, the first term is $dx/|\omega^{(D)}|$ with 2-cocycle $x$ whose value on face $f$ is as follows: denote the value of $B_2$ on face $f$ by $g = B_2(f)$, then $x(f) = i^B_g(\omega^{(D)})$, which gives an element in $H^{D-2}(BG, U(1))$. The second term computes the cup product of face $f$, which has $g = B_2(f)$, with edge $e$ in the 3-simplex, which has $C_1(e) \in H^{D-1}(BG, U(1))$, with the cup product coefficient given by $i^A_g(C_1(e)) \in H^{D-2}(BG, U(1))$.

The second term on the rhs above describes the codimension-2 defect that arises from dimensionally reducing the codimension-1 gauged SPT defect, described by $C_1$, on the magnetic defect described by $B_2$. By introducing a coboundary operation $\delta_\rho$ twisted by $\rho$, we can rewrite the above equation as

$$d_\rho C_2 = -d(i^B_{B_2}\omega^{(D)})/|\omega^{(D)}| \tag{92}$$

The equation is the twisted version of (28) in the presence of the background gauge field $C_1$, and it describes the action of the 0-form symmetry acting on the total 1-form symmetry given by the extension of the center 1-form symmetry $Z_\omega(G)$ by $H^{D-2}(BG, U(1))$.

In this case, the 0-form and 1-form symmetry groups $\mathcal{G}^{(0)}$ and $\mathcal{G}^{(1)}$ describe a 2-group symmetry with a trivial $H^3(B\mathcal{G}^{(0)}, \mathcal{G}^{(1)})$ Postnikov class. It is possible to get a non-trivial $H^3$ Postnikov class by considering the permutation action arising from $S \subset \mathcal{G}^{(0)}$. Some examples of such non-trivial classes were studied in (2+1)D finite gauge theories based on dihedral groups [23, 43].

### 6.2.2   $n = 3$, 3-group symmetry, and $H^4$ Postnikov class

We have

$$f_4 = -i^A_{B_2}(C_2) - i^A_{B_2}i^A_{B_2}(C_1) - i^B_{\text{Bock}(B_2)}(C_1) - i^B_{B_2}(\text{Bock}(C_1)) - i^A_{B_2}d(i^B_{B_2}\omega^{(D)})/|\omega^{(D)}| \ . \tag{93}$$

On a 4-simplex, the first term receives contributions from cup product of surfaces $f, f'$, where $B_2(f) = g \in Z_\omega(G)$, and $C_2(f') \in H^{D-2}(BG, U(1))$, and the coefficient of the cup product is given by $i^A_g C_2(f')$, which is an element in $H^{D-3}(BG, U(1))$. The second term receives contribution from faces $f, f'$ and edge $e$ where $(\tilde{f} \cup_1 \tilde{f'}) \cup \tilde{e} \neq 0$ where $\tilde{f}, \tilde{f'}, \tilde{e}$ denote the cochains that takes value 1 on $f, f', e$ and 0 otherwise. We have $B_2(f) = g, B_2(f') = g' \in Z_\omega(G)$, and $C_1(e) \in H^{D-1}(BG, U(1))$. The coefficient of $(\tilde{f} \cup_1 \tilde{f'}) \cup \tilde{e}$ is given by $i^A_g i^A_{g'} C_1(e) \in H^{D-3}(BG, U(1))$. The remaining terms receive contribution from 3-simplex $c$ and edge $e$ such that $\tilde{c} \cup \tilde{e} \neq 0$, from faces $f, f'$ such that $\tilde{f} \cup \tilde{f'} \neq 0$, and from face $f$ and 3-simplex $c$ such that $\tilde{f} \cup_1 \tilde{c} \neq 0$.

Each term on the right hand side above can be understood physically as follows. The first term, $-i^A_{B_2}(C_2)$, describes a codimension-3 gauged SPT defect (described by $C_3$) that is sourced by the codimension-3 crossing between the codimension-2 magnetic defect (described by $B_2$) and the codimension-2 gauged SPT defect (described by $C_2$). In the case $D = 4$, this corresponds to an electric charge that is sourced at the crossing point between the codimension-2 magnetic and gauged SPT defects, and was analyzed in [83]. Using $f_3$, we find that the first term and the last term combine into a term of the form

$$B_2 \cup C_2 + B_2 \cup_1 dC_2 \ . \tag{94}$$

The second term can be rewritten as the following form:[19]

$$-i^A_{B_2}i^A_{B_2}(C_1) = -(B_2 \cup_1 B_2) \cup C_1. \tag{95}$$

The physical meaning of this term was analyzed in [83] in the context of (3+1)D untwisted $\mathbb{Z}_2^3$ gauge theory. It describes a source for a codimension-3 gauged SPT defect arising from the intersection of two magnetic defects on the codimension-1 gauged SPT domain wall described by $C_1$. In the case $D = 4$, this corresponds to a source of electric charge at the (non-generic) crossing point between the magnetic defects and the codimension-1 gauged SPT domain wall.

---

[19]We note that such term produces the shift in $C_3$ by $\lambda \cup \lambda \cup C_1$ using $(d\lambda \cup_1 d\lambda) \cup C_1 = d((\lambda \cup \lambda - \lambda \cup_1 d\lambda) \cup C_1)$.

The third term, $-i^B_{\text{Bock}(B_2)}(C_1)$, arose in our example in Section 3.2 on (3+1)D untwisted $\mathbb{Z}_2^2$ gauge theory. It describes how a junction of magnetic defects, intersecting with a codimension-1 gauged SPT domain wall, sources a codimension-3 gauged SPT defect. In the case of $D = 4$, this corresponds to a source for an electric charge, as seen in Figure 6.

The fourth term, $-i^B_{B_2}(\text{Bock}(C_1))$, arose in our example in Section 3.1 on (3+1)D untwisted $\mathbb{Z}_2$ gauge theory. The intersection of the magnetic defect with the codimension-2 junction of codimension-1 domain walls, sources a codimension-3 gauged SPT defect. In the case $D = 4$, this corresponds to a source of electric charge, as shown in Figure 3.

Finally, the fifth term, $-i^A_{B_2} i^B_{dB_2/|\omega^{(D)}|} \omega^{(D)}$, can be understood as a combination of effects described previously. First, the codimension-3 junction of magnetic defects can 'emit' a codimension-2 gauged SPT defect, as described in Section 2.3. Next, the codimension-3 intersection between the codimension-2 magnetic defect described by $B_2$ and the codimension-2 emitted SPT, described by $i^B_{dB_2/|\omega^{(D)}|} \omega^{(D)}$, sources a codimension-3 gauged SPT defect. In the case $D = 4$, this corresponds to yet another possible source of electric charge.

The combined terms in $f_4$ can all be understood as a Postnikov class $[f_4] \in H^4(B\mathbb{G}^{(2)}, \mathcal{G}^{(2)})$. The background gauge fields $\{C_1, C_2, B_2\}$ define a flat background gauge field for the 2-group $\mathbb{G}^{(2)}$. Altogether, $\{\mathbb{G}^{(2)}, \mathcal{G}^{(2)}, [f_4]\}$ define the 3-group symmetry $\mathbb{G}^{(3)}$.

## 6.3 Example: untwisted $\mathbb{Z}_2^3$ gauge theory in (3+1)D

Let us illustrate the previous discussion using twisted $\mathbb{Z}_2^3$ gauge theory in (3+1)D, which has 3-group symmetry as discussed in [83]. In Appendix B we give another derivation of the 3-group symmetry by embedding the discrete gauge fields into continuous $U(1)$ gauge fields.

Denote the $\mathbb{Z}_2$ gauge fields by $a_1, a_2, a_3$, and the shifts by $\lambda_1, \lambda_2, \lambda_3$. Let us consider the 0-form symmetry generated by $(-1)^{\int a_1 \cup a_2 \cup a_3}$, and the 1-form symmetry generated by the gauged SPT phase $(-1)^{\int a_i \cup a_j}$ for $i < j$, and the 2-form symmetry generated by the Wilson lines. Denote their backgrounds by $C_1, C_2^{ij} = C_2^{ji}, C_3^i$. Let us consider the action

$$\int \left( C_1 \cup a_1 \cup a_2 \cup a_3 + C_2^{12} \cup a_1 \cup a_2 + C_2^{13} \cup a_1 \cup a_3 + C_2^{23} \cup a_2 \cup a_3 + \sum_i C_3^i \cup a_i \, . \right) \tag{96}$$

Under the shift $a_i \to a_i + \lambda_i$ with $d\lambda_i = 0$, for the action to be invariant up to classical terms, the backgrounds are shifted as

$$C_2^{ij} \to C_2^{ij} + \epsilon^{ijk} C_1 \cup \lambda_k,$$
$$C_3^i \to C_3^i + (C_2^{il} + \epsilon^{ilm} C_1 \cup \lambda_m) \cup \lambda_l + C_1 \cup (\frac{1}{2}\epsilon^{ijk}\lambda_j \cup \lambda_k) = C_3^i + C_2^{il} \cup \lambda_l \, . \tag{97}$$

This is consistent with the relations derived in [83],

$$dC_2^{ij} = \epsilon^{ijk} C_1 \cup B_2^k, \quad dC_3^i = C_2^{ij} \cup B_2^j + C_1 \epsilon^{ijk} \cup (B_2^j \cup_1 B_2^k) \, , \tag{98}$$

where $B_2^i$ is the background for the center 1-form symmetry that transforms $a_i$. In the above relation between background gauge fields, $B_2^k \to B_2^k + d\lambda_k$ reproduces (97) in the case of trivial backgrounds $B_2^k$ and closed $\lambda_k$.

# 7 Anomaly of $d$-group symmetry in Dijkgraaf-Witten gauge theory

The anomaly of $d$-group symmetry $\mathbb{G}^{(d)}$ in spacetime dimension $D$ can be described by an SPT phase in a $(D+1)$-dimensional bulk with the $d$-group symmetry. Such SPT phases are characterized by an element in $H^{D+1}(B\mathbb{G}^{(d)}, U(1))$ [41] (recall $D = d + 1$), and can be described in terms of a bulk effective action that depends on the background gauge field for the $d$-group symmetry, extended into the $(D+1)$-dimensional space-time. In the following, we will derive the anomaly by studying the correlation functions of the symmetry generators.

The correlation function is non-trivial only when the magnetic defect is present. For magnetic defect carrying holonomy $g$ in the center of gauge group, and supported on $M_{D-2}^g$, which is on the boundary of $V_{D-1}^g$, the defect sources the gauge field

$$a = \sum_{g \in Z_\omega(G)} a_g, \quad a_g \equiv g\delta(V_{D-1}^g)^\perp \, . \tag{99}$$

Then the correlation function of magnetic defect $U_g$ and gauged SPT defect $\mathcal{W}_{\eta^{(n)}} = e^{i \int a^* \eta^{(n)}}$ is given by[20]

$$\left\langle \left( \prod_{g \in Z_\omega(G)} U_g(M_{D-2}^g) \right) \prod_n \mathcal{W}_{\eta^{(n)}}(M_n) \right\rangle = \prod_{g \in Z_\omega(G)} e^{i \int_{V_{D-1}^g} a_g^*(i_g \omega^{(D)})} \prod_n e^{i \int_{M_n} a_g^*(\eta^{(n)})} . \tag{100}$$

For instance, consider $\omega^{(D)} = 0$, and the 0-cocycle $i_{g_1} i_{g_2} \cdots i_{g_n} \eta^{(n)} \neq 0$ is a nonzero constant, and take $M_n$ to be an $n$-dimensional torus. The correlation function of the gauged SPT defect $\mathcal{W}_{\eta^{(n)}}(M_n)$ and the magnetic defects of holonomies $g_1, \cdots, g_{n-1}, g_n$ is given by $(n+1)$-linking number:

$$\left\langle U_{g_1}(M_{D-2}^{g_1}) \cdots U_{g_n}(M_{D-2}^{g_n}) \mathcal{W}_{\eta^{(n)}}(M_n) \right\rangle = e^{i \left( i_{g_1} i_{g_2} \cdots i_{g_n} \eta^{(n)} \right) \text{Link}\left( \{ M_{D-2}^{g_i} \}_{i=1}^n, M_n \right)} ,$$

$$\text{Link}\left( \{ M_{D-2}^{g_i} \}_{i=1}^n, M_n \right) \equiv \int \delta(V_{D-1}^{g_1})^\perp \cup \delta(V_{D-1}^{g_2})^\perp \cdots \cup \delta(V_{D-1}^{g_n})^\perp \cup \delta(M_n)^\perp , \tag{101}$$

where the integral in $\text{Link}\left( \{ M_{D-2}^{g_i} \}_{i=1}^n, M_n \right)$ is over the $D$-dimensional spacetime. Examples with 3-loop braiding in (3+1)D [135] described by the above correlation functions are discussed in [136, 44].

**'t Hooft anomaly of $d$-group symmetry**    In terms of background gauge fields

$$n \neq 2: \quad C_{D-n} = \delta(M_n)^\perp, \quad B_2 = d\lambda, \quad \lambda \equiv \sum_g g\delta(V_{D-1}^g)^\perp, \quad C_2 = \delta(M_{D-2}')^\perp , \tag{102}$$

the correlation function is given by $e^{iS_{\text{bulk}}}$, where on a coordinate patch we have

$$S_{\text{bulk}} = \int_{\text{bulk}} \sum_g \left( d(a_g^*(i_g \omega^{(D)}) \cup \delta(V_{D-1}^g)^\perp) + \sum_n \left( da_g^*(\eta^{(D-n)}) \cup C_n + (-1)^{D-n} a_g^*(\eta^{(D-n)}) \cup dC_n \right) \right) . \tag{103}$$

This is the bulk term contributed from a coordinate patch, where $B_2 = d\lambda$ is exact. The bulk integral is given by summing over the contribution from all coordinate patches, with transition function that changes $\lambda$ to produce non-trivial $B_2$ background gauge field by the analogue of the clutching construction for vector bundles.

# 8    3-group symmetry in $\mathbb{Z}_2$ gauge theory with fermion particles in (3+1)D

In this section we discuss the 3-group symmetry in bosonic $\mathbb{Z}_2$ gauge theory in (3+1)D where the electric particles are emergent fermions. Such theory in the absence of magnetic defects can be described by a $\mathbb{Z}_2$ gauge field $a$ that satisfies $da = w_2(TM)$, where $w_2(TM)$ is the second Stiefel-Whitney class of the spacetime manifold [137]. We do not include additional topological action for $a$ in the path integral.

As an application and consistency check, we use the 3-group symmetry to enrich the theory with ordinary symmetry $G_b$ (in this section, $G_b$ will be a global symmetry instead of the gauge group of dynamical gauge field), by embedding the $G_b$ bundle into the 3-group bundle, following the discussion in [41]. The resulting theory is the 'bosonic shadow' of a fermionic SPT phase with $G_b$ and $\mathbb{Z}_2^f$ symmetries, related by gauging the fermion parity symmetry $\mathbb{Z}_2^f$. We show that the 3-group symmetry in the $\mathbb{Z}_2$ gauge theory with fermionic electric charge provides a compact and explicit way to characterize and classify interacting fermionic SPT phases in (3+1)D based on field theory. This gives a simpler approach as compared with the results of [34], which is based on explicit microscopic constructions of fixed point wave functions. In particular, we obtain a simple expression for the $O_5$ obstruction, and we show that it agrees with [34]. Ref. [138] followed a similar approach, but considered a subclass of fermion SPT phases (those described by group supercohomology [139]) and restricted to the case where the fermionic symmetry group splits as $G_f = G_b \times \mathbb{Z}_2^f$; our approach applies to general fermionic SPT phases with general unitary internal $G_f$ symmetry.

## 8.1    3-group symmetry

Since the gauge theory has fermionic particles, the invertible topological defects are obtained by gauging the fermion parity symmetry on submanifolds decorated with invertible topological phases. The theory has the following symmetries:

---

[20]The pullback of $\eta^{(n)}$ by $a_g$ can be computed on any $n$-simplex, where each edge has the group element given by the field configuration $a_g$, such that the gauge field has the prescribed flux specified by the magnetic defects.

- 2-form symmetry generated by the Wilson line

$$W(\gamma) = (-1)^{\oint_\gamma a} \tag{104}$$

- 1-form symmetry generated by the magnetic surface defect $V(\Sigma)$.

- 1-form symmetry generated by the gauged invertible fermionic phases in $(1+1)$D: on surface $\Sigma$, the defect is [31, 53]

$$U(\Sigma) = \exp\left(\frac{\pi i}{2} \int_\Sigma q_\rho(a')\right) , \tag{105}$$

where $\rho$ is the spin structure on the defect, $q_\rho$ is the quadratic function on $\Sigma$ that takes $\mathbb{Z}_2$ valued 1-form and produces a $\mathbb{Z}_4$ valued 2-form [140, 141], and $a'$ is a dynamical $\mathbb{Z}_2$ gauge field related to the $(3+1)$D gauge field by

$$a|_\Sigma = a' + \rho , \tag{106}$$

where $a|_\Sigma$ is the restriction of $a$ to the defect. In particular, in the absence of the magnetic defects, $da' = 0$. We note that the topological action $\frac{\pi}{2} \int q_\rho(a')$ describes the Kitaev chain [142, 143] as discussed in [31, 144, 53].

To investigate the 3-group symmetry, let us couple $U$ to a background 2-form gauge field $C_2$, and we turn on background gauge field $B_2$ that couples to the center 1-form symmetry generated by the magnetic defect, and background $C_3$ for the 2-form symmetry generated by the Wilson line:

$$S = \frac{\pi}{2} \int q_\rho(a') \cup C_2 + \pi \int a \cup C_3, \quad da = w_2(TM) + B_2 , \tag{107}$$

where the Wilson line of $a$ describes bulk emergent fermion particle, and the first term can be written as an integral over the Poincaré dual of $C_2$, where the spin structure $\rho$ and the gauge field $a'$ is well-defined.

To see the 3-group symmetry, let us follow the procedure in Section 6. Let us perform a transformation of the 1-form center symmetry (whose background is $B_2$)

$$a \to a + \lambda, \quad B_2 \to B_2 + d\lambda . \tag{108}$$

We will take 1-form $\lambda$ to satisfy $d\lambda = 0$, and thus we can turn off $B_2$. Using the property of quadratic function $q(x+y) = q(x)+q(y)+2x\cup y$ for $\mathbb{Z}_2$ gauge fields $x, y$, we find that for the action to be invariant up to classical terms that only depend on the backgrounds $C_2, C_3, \lambda$ but not the dynamical fields, the backgrounds need to be shifted as

$$C_3 \to C_3 + \lambda \cup C_2 . \tag{109}$$

This implies that if we insert a magnetic defect that intersects with the surface supporting the defect $U$, there is additional electric charge. This is consistent with [83].

Moreover, if we perform transformation $C_2 \to C_2 + d\lambda'$ (and keep $B_2 = 0$), the first term in (107) changes by

$$\frac{\pi}{2} \int q_\rho(a') \cup d\lambda' = \pi \int a' \cup w_2(TM) \cup \lambda' , \tag{110}$$

where the equality follows from integration by parts and the dependence of the quadratic function on the spin structure [140, 141, 53]. Thus to compensate for the change under the transformation $\lambda'$, the background $C_3$ is shifted as

$$C_3 \to C_3 + w_2(TM) \cup \lambda' . \tag{111}$$

We note that the surface defect $U$ that couples to the background $C_2$ squares to the Wilson line operator $W$ that couples to the background $C_3$, when the surface that supports the defect has self-intersections:

$$(U^{(2)}(\Sigma))^2 = (-1)^{\oint_\Sigma a' \cup a'} = (-1)^{\oint_\Sigma a' \cup w_1(T\Sigma)} = (-1)^{\oint_\gamma a'} = W(\gamma) , \tag{112}$$

where we used the Wu formula on $\Sigma$, $a' \cup a' = w_1(T\Sigma) \cup a'$ [137], and $w_1(T\Sigma) = \delta_\Sigma(\gamma)^\perp$ where the subscript in $\delta_\Sigma$ restricts the perpendicular direction with respect to embedding the curve $\gamma$ inside $\Sigma$ to be zero.[21] The property that the surface defect squares to the line defect implies that the 1-form symmetry combines with the 2-form symmetry into a 3-group.

---

[21] Alternatively, we can write the exponent as an integral over the spacetime $\int a' \cup a' \cup \delta(\Sigma)^\perp = \int Sq^1(a') \cup \delta(\Sigma)^\perp = \int a' \cup \delta(\Sigma)^\perp$, and thus $\int_\Sigma a' \cup a'$ is equivalent to the Wilson line $\int a'$ inserted at the Poincaré dual of $Sq^1\delta(\Sigma)^\perp$.

We can summarize the transformations of $C_3$ in (109) and (111) into the following relation between the background gauge fields

$$dC_3 = w_2(TM) \cup C_2 + B_2 \cup C_2 = Sq^2(C_2) + B_2 \cup C_2 \ , \tag{113}$$

and the Postnikov class of the 3-group symmetry is

$$f_4(B_2, C_2) = Sq^2(C_2) + B_2 \cup C_2 \ . \tag{114}$$

This is consistent with [83].

**'t Hooft anomaly**   Let us compute the 't Hooft anomaly using the correlation function of the symmetry generators. If we insert the magnetic defect $V(M_2)$ at surface $M_2$, which is on the boundary of $V_3$, it activates the gauge field

$$a = a_g, \quad a_g = \delta(V_3)^{\perp} \ . \tag{115}$$

The correlation function is given by substituting the above gauge flield into the expression for the gauged SPT defects:

$$\langle U(\Sigma)V(M_2)W(\gamma)\rangle = e^{\pi i \int_{\gamma} \delta(V_3)^{\perp} + \frac{\pi i}{2} \int_{\Sigma} q_{\rho}(\delta(V_3)^{\perp} + \rho)} \ . \tag{116}$$

The configuration of the above symmetry generators is equivalent to the background gauge fields

$$C_2 = \delta(M_2)^{\perp}, \quad C_3 = \delta(\gamma)^{\perp}, \quad B_2 = w_2(TM) + \delta(M_2)^{\perp} = w_2(TM) + d\delta(V_3)^{\perp} \ , \tag{117}$$

where there is a shift in $B_2 = w_2(TM) + \delta(M_2)^{\perp}$ by $w_2(TM)$, since the Wilson line is a fermion.

The correlation function can be written in terms of the background gauge fields as $e^{iS_{\text{bulk}}}$, where

$$S_{\text{bulk}} = \pi \int_Y C_3 \cup (B_2 - w_2(TM)) + \frac{\pi}{2} \int_Y \left( dq_{\rho}(\delta(V_3)^{\perp} + \rho)C_2 + q_{\rho}(\delta(V_3)^{\perp} + \rho)dC_2 \right) \ , \tag{118}$$

where $Y$ is a (4+1)D manifold that bounds the spacetime. This is the expression of the effective action on a coordinate patch for the bulk invertible phase with 3-group symmetry that describes the anomaly, where $B_2 = d\lambda$ with $\lambda = \delta(V_3)^{\perp}$. The effective action on general bulk manifold can then be obtained by the analogue of clutching construction for vector bundles. In particular, the first term in the bulk effective action is

$$S_{\text{bulk}} \supset \pi \int C_3 \cup (B_2 - w_2(TM)) = \pi \int \left( C_3 \cup B_2 + Sq^2(C_3) \right) \ , \tag{119}$$

where the equality uses the Wu formula [137].

## 8.2   Application: explicit description for fermionic SPT phases in (3+1)D

### 8.2.1   Review of fermionic SPT phases in (3+1)D

Let us consider fermionic SPT phases in (3+1)D with an internal 0-form fermionic symmetry group $G_f$. $G_f$ is in general a central extension of the bosonic symmetry group $G_b$ by $\mathbb{Z}_2^f$, specified by extension class $[\omega_2] \in H^2(BG_b, \mathbb{Z}_2)$. Such fermionic SPTs are known to be classified by the following data: [34] $n_1 \in C^1(BG_b, \mathbb{Z}^T), n_2 \in C^2(BG_b, \mathbb{Z}_2), n_3 \in C^3(BG_b, \mathbb{Z}_2), \nu_4 \in C^4(BG_b, U(1))$, satisfying

$$
\begin{aligned}
dn_1 &= 0 \ , \\
dn_2 &= \omega_2 \cup n_1 + s_1 \cup n_1 \cup n_1 \ , \\
dn_3 &= (n_2 + \omega_2) \cup n_2 + s_1 \cup (n_2 \cup_1 n_2) \ . \\
d\nu_4 &= O_5
\end{aligned}
\tag{120}
$$

Here $s_1 \in Z^1(BG_b, \mathbb{Z}_2)$ and $s_1(g) \neq 0$ means that $g$ is an anti-unitary symmetry. $\mathbb{Z}^T$ in $C^1(BG_b, \mathbb{Z}^T)$ means that $G_b$ acts on $\mathbb{Z}$ through the map $s_1$. We note that $n_1$ modifies the cocycle conditions in the above equations only for non-trivial $s_1$. The formula of $O_5$ is given in [34]. The physical meaning of $n_1$ is the decoration of the p+ip topological superconductor on the domain wall that generated the time-reversal symmetry.

We will focus on the case of unitary symmetry $G_b$. In such a case, $n_1 = 0, s_1 = 0$, and we have

$$dn_1 = 0 \ , \qquad dn_2 = 0 \ , \qquad dn_3 = (n_2 + \omega_2) \cup n_2 \ , \qquad d\nu_4 = O_5 \ , \tag{121}$$

where $O_5$ is given in Eq. (220) of Ref. [34].

In the following, we will reproduce (121) using the 3-group symmetry in the bosonic shadow theory of the fermionic SPT phases, which are $\mathbb{Z}_2$ gauge theories with fermionic particles enriched by 0-form symmetry $G_b$.

### 8.2.2 Fermionic SPT phases from bosonic shadow with 3-group symmetry

Following the approach in [35], we study the fermionic SPT phases using their bosonic shadow obtained by gauging the fermion parity symmetry. Such bosonic shadow theory is the $\mathbb{Z}_2$ gauge theory with fermion electric charge in (3+1)D, and it can be described by (3+1)D fermionic $\mathbb{Z}_2$ Toric code model. Let us discuss the meaning of the classifying data $n_1, n_2, n_3, \nu_4$ for fermionic SPT phases in this $\mathbb{Z}_2$ gauge theory.

In the (3+1)D fermionic toric code, $n_2$ corresponds to the decoration of codimension-two defect (gauged Kitaev chain), $\omega_2$ corresponds to the $m$-loop, and $n_3$ corresponds to the fermion $\psi$. In other words, denote the background $G_b$ gauge field by $A$, this corresponds to

$$C_3 = A^* n_3, \quad C_2 = A^* n_2, \quad B_2 = A^* \omega_2 . \tag{122}$$

Then the 3-group relation (113) reproduces the constraint on the classification data $n_3, n_2, \omega_2$ in (121). The obstruction $O_5$ is given by the anomaly (118) of the 3-group symmetry. In particular, the first term (119) in the anomaly gives

$$d\nu_4 = O_5, \quad O_5 = \frac{1}{2}(n_3 \cup_1 n_3 + n_3 \cup_2 dn_3 + \omega_2 \cup n_3) + \cdots . \tag{123}$$

We can then complete the rest by demanding the anomaly to be closed, and we summarize the computation details in Appendix C. We obtain

$$O_5 \equiv \frac{1}{2} \left( n_3 \cup_1 n_3 + n_3 \cup_2 dn_3 + \omega_2 \cup n_3 + dn_3 \cup_1 n_2 + \zeta_5(n_2 + \omega_2, n_2) \right) + \frac{1}{4}[n_2 + \omega_2]_2 \cup (n_2 \cup_1 n_2) . \tag{124}$$

In Appendix C we show that this expression of $O_5$ agrees with Eq. (220) of Ref. [34], but is significantly simpler.

## Acknowledgement

We thank Guanyu Zhu for helpful conversations. We thank Xie Chen, Meng Cheng, Arpit Dua, Chao-Ming Jian and Shu-Heng Shao for their comments on the draft. MB acknowledges financial support from NSF CAREER DMR- 1753240. YC and RK are supported by the JQI postdoctoral fellowship at the University of Maryland and by the Laboratory for Physical Sciences through the Condensed Matter Theory Center. P.-S. H. is supported by the Simons Collaboration on Global Categorical Symmetries. The work is performed in part at Aspen Center for Physics, which is supported by National Science Foundation grant PHY-1607611. The authors thank the American Institute of Mathematics for hosting the workshop "higher categories and topological order" where this work was partially performed.

## A Properties of $n$-group symmetry

An $n$-group symmetry $\mathbb{G}^{(n)}$ is an extension of $q$-form symmetries for $q < n - 1$ by $(n-1)$-form symmetry $\mathcal{G}^{(n-1)}$. The classifying space of an $n$-group $\mathbb{G}^{(n)}$ is a fibration of $B^n \mathcal{G}^{(n-1)}$ over the classifying space of an $(n-1)$-group:

$$B^2 \mathcal{G}^{(1)} \to B\mathbb{G}^{(2)} \to B\mathcal{G}^{(0)}, \quad B^3 \mathcal{G}^{(2)} \to B\mathbb{G}^{(3)} \to B\mathbb{G}^{(2)}, \quad \cdots, \quad B^n \mathcal{G}^{(n-1)} \to B\mathbb{G}^{(n)} \to B\mathbb{G}^{(n-1)} . \tag{125}$$

The background gauge fields for an $n$-group $\mathbb{G}$ consists of the background gauge field $C_n$ for the top $(n-1)$-form symmetry, and the backgrounds for the lower $(k-1)$-form symmetries $C_k$ with $1 \le k \le n-1$, and they obey the constraints

$$dC_1 = 0, \quad dC_k = f_{k+1}(\{C_j\}_{j=1}^{k-1}) \quad \text{for } k \ge 2 , \tag{126}$$

where $[f_{k+1}] \in H^{k+1}(B\mathbb{G}^{(k-1)}, \mathcal{G}^{(k-1)})$ are called the Postnikov classes. The relations between the background gauge fields can equivalently be described by the relation between the background gauge transformations: for the transformation parameters $\lambda_k$ of degree $k$,

$$C_1 \to \lambda_0^{-1} C_1 \lambda_0 + \lambda_0^{-1} d\lambda_0, \quad C_2 \to C_2 + \alpha_2(C_1, \lambda_0) + d\lambda_1, \quad C_3 \to C_3 + \alpha_3(C_1, C_2, \lambda_0, \lambda_1) + d\lambda_2, \quad \cdots , \tag{127}$$

where $\alpha_n$ are the anomaly descendants of $f_{n+1}$.

# B  Higher-group symmetry in Abelian gauge theories in the continuum

Let us give another derivation of the higher-group symmetry in the examples in Section 3 by embedding the discrete gauge fields into continuous $U(1)$ gauge fields. In other words, a $\mathbb{Z}_N$ 1-form gauge field can be described by a pair of $U(1)$ gauge fields $a, b$ [145, 146]

$$\frac{N}{2\pi} a db \ , \tag{128}$$

where $a$ is a $U(1)$ 1-form gauge field, and $b$ is a $(D-2)$-form $U(1)$ gauge field that plays the role of the Lagrangian multiplier constraining $a$ to have $\mathbb{Z}_N$ holonomy. As described in Ref. [146], $b$ can also be dualized into a Higgs field that breaks the gauge group of $a$ from $U(1)$ to $\mathbb{Z}_N$. The operator $e^{i \oint b_2}$ has $e^{2\pi i/N}$ braiding with the Wilson line $e^{i \oint a}$, and it is the magnetic defect of codimension two that carries unit holonomy of $\mathbb{Z}_N$ gauge field.

## B.1  Example: $\mathbb{Z}_N$ gauge theory in (3+1)D

Let us turn on the background gauge fields $C_1$ of the 0-form symmetry generated by the domain wall $e^{\frac{i}{2\pi} \int ada}$ decorated with gauged Levin-Gu phase, background $B_2$ of the center 1-form symmetry generated by the magnetic defect $e^{i \oint b}$, and background $C_3$ for the 2-form symmetry generated by the Wilson line $e^{i \int a}$. The action is

$$\int \left( \frac{N}{2\pi} a db + \frac{1}{2\pi} ada C_1 + b B_2 + a C_3 \right) \ . \tag{129}$$

The equations of motion for $b, a$ are

$$\frac{N}{2\pi} da + B_2 = 0, \quad \frac{N}{2\pi} db + \frac{1}{2\pi}(2da C_1 - a dC_1) + C_3 = 0 = \frac{N}{2\pi} db + \frac{1}{2\pi}(-\frac{4\pi}{N} B_2 C_1 - a dC_1) + C_3 \ . \tag{130}$$

Taking the differential of the second equation gives

$$dC_3 - \frac{1}{N} B_2 dC_1 - \frac{2}{N} dB_2 C_1 = 0 \ . \tag{131}$$

For $N = 2$, this reproduces Eq. (35).

Let us also derive the symmetry by studying the 1-form transformation $a \to a - \frac{2\pi}{N}\lambda, \quad B_2 \to B_2 + d\lambda$. For the action to be invariant, the background $C_3$ transforms by the shift

$$C_3 \to C_3 + \frac{1}{N} d\lambda C_1 \ , \tag{132}$$

up to a total derivative (which is the usual background gauge transformation of $C_3$). The is the transformation in Eq. (131) that leaves the relation invariant.

## B.2  Example: untwisted $\mathbb{Z}_N^2$ gauge theory in (3+1)D

Let us denote the two $\mathbb{Z}_N$ gauge fields by $a, a'$, and consider the background $C_1$ for the 0-form symmetry generated by $e^{\frac{i}{2\pi} \int ada'}$, the backgrounds $B_2, B_2'$ for the center 1-form symmetry generated by the magnetic defects $e^{i \int b}, e^{i \int b'}$, the backgrounds $C_3, C_3'$ for the 2-form symmetry generated by the Wilson lines $e^{i \int a}, e^{i \int a'}$. The action is

$$\int \left( \frac{N}{2\pi} a db + \frac{N}{2\pi} a' db' + \frac{1}{2\pi} ada' C_1 + b B_2 + b' B_2' + a C_3 + a' C_3' \right) \ . \tag{133}$$

The equations of motion of $b, b', a, a'$ give

$$\begin{aligned}
&\frac{N}{2\pi} da + B_2 = 0, \quad \frac{N}{2\pi} da' + B_2' = 0 \\
&\frac{N}{2\pi} db + \frac{1}{2\pi} da' C_1 + C_3 = 0 = \frac{N}{2\pi} db - \frac{1}{N} B_2' C_1 + C_3, \\
&\frac{N}{2\pi} db' + \frac{1}{2\pi} d(a C_1) + C_3' = 0 = \frac{N}{2\pi} db' - \frac{1}{N} B_2 C_1 - \frac{1}{2\pi} a dC_1 + C_3' \ .
\end{aligned} \tag{134}$$

Taking the differential of the last two lines gives

$$dC_3 - \frac{1}{N} dB_2' C_1 - \frac{1}{N} B_2' dC_1 = 0, \quad dC_3' - \frac{1}{N} dB_2 C_1 = 0 \ . \tag{135}$$

This reproduces Eq. (48).

Let us also derive the symmetry by studying the 1-form transformation $a \to a - \frac{2\pi}{N}\lambda, \quad B_2 \to B_2 + d\lambda, \quad a' \to a' - \frac{2\pi}{N}\lambda', B_2' \to B_2' + d\lambda'$. For the action to be invariant, the background $C_3, C_3'$ transforms by the shift

$$C_3 \to C_3 + \frac{1}{N} d\lambda' C_1 \, , \tag{136}$$

up to a total derivative (which is the usual background gauge transformation of $B_3, B_3'$). The is the transformation in Eq. (135) that leaves the relations invariant.

## B.3  Example: untwisted $\mathbb{Z}_N^3$ gauge theory in (3+1)D

Similarly, let us consider $\mathbb{Z}_N^3$ gauge theory with gauge fields $a_1, a_2, a_3$. Let us turn on backgrounds $C_3^i, C_2^{ij}, B_2^i, C_1$ with $i, j = 1, 2, 3$ labeling the three gauge fields. The action is

$$\int \left( \frac{N}{2\pi} (a_1 db_1 + a_2 db_2 + a_3 db_3) + \frac{N}{2\pi} \left( a_1 a_2 C_2^{12} + a_3 a_1 C_2^{31} + a_2 a_3 C_2^{23} \right) \right)$$
$$+ \int \left( \frac{N^2}{(2\pi)^2} a_1 a_2 a_3 B_1 + b_1 B_2^1 + b_2 B_2^2 + b_3 B_2^3 + a_1 C_3^1 + a_2 C_3^2 + a_3 C_3^3 \right) \, . \tag{137}$$

The equation of motions are

$$\frac{N}{2\pi} da_i + B_2^i = 0$$
$$\frac{N}{2\pi} db_1 + \frac{N}{2\pi} \left( a_2 C_2^{12} - a_3 C_2^{31} \right) + \frac{N^2}{(2\pi)^2} a_2 a_3 C_1 + C_3^1 = 0$$
$$\frac{N}{2\pi} db_2 + \frac{N}{2\pi} \left( -a_1 C_2^{12} + a_3 C_2^{23} \right) + \frac{N^2}{(2\pi)^2} a_3 a_1 C_1 + C_3^2 = 0$$
$$\frac{N}{2\pi} db_3 + \frac{N}{2\pi} \left( -a_2 C_2^{23} + a_1 C_2^{31} \right) + \frac{N^2}{(2\pi)^2} a_1 a_2 C_1 + C_3^3 = 0 \, . \tag{138}$$

By taking the differentials of the above equations and comparing the coefficients of $a_i, a_i a_j$, we find

$$dC_2^{12} - B_2^3 C_1 = 0, \quad dC_2^{31} - B_2^2 C_1 = 0, \quad dC^{23} - C_2^1 B_1 = 0$$
$$dC_3^1 - B_2^2 C_2^{12} + B_2^3 C_2^{31} = 0, \quad dC_3^2 + B_2^1 C_2^{12} - B_2^3 C_2^{23} = 0, \quad dC_3^3 + B_2^2 C_2^{23} - B_2^1 C_2^{31} = 0 \, , \tag{139}$$

This reproduces the relation (98) without the $\cup_1$ product, since for the background gauge fields that can be embedded into $U(1)$ background gauge fields, they satisfy super-commutativity as in differential forms, and the contributions from higher cup products are trivial.

# C  Computation detail for fermionic SPT phases

In this appendix, we give the details for discussing the fermionic SPT phases in (3+1)D in Section 8.2, where we used the 3-group symmetry to classify the fermionic SPT phases.

A useful formula is the Cartan formula [147]

$$Sq^2(A_2 \cup B_2) \overset{2}{=} Sq^2 A_2 \cup B_2 + Sq^1 A_2 \cup Sq^1 B_2 + A_2 \cup Sq^2 B_2 + d\zeta_5(A_2, B_2) \, , \tag{140}$$

with $A_2, B_2 \in Z^2(M, \mathbb{Z}_2)$ and[22]

$$\zeta_5(A_2, B_2)(012345) \equiv A_2(023) A_2(012) B_2(345) B_2(235). \tag{141}$$

We choose $A_2 = n_2 + \omega_2$ and $B_2 = n_2$ and the Cartan formula gives

$$Sq^2(dn_3) \overset{2}{=} Sq^2(n_2 + \omega_2) \cup n_2 + Sq^1(n_2 + \omega_2) \cup Sq^1 n_2 + (n_2 + \omega_2) \cup Sq^2 n_2 + d\zeta(n_2 + \omega_2, n_2) \, . \tag{142}$$

---

[22]If we define $\bar{A}(ij) = A(0ij)$, the above $\zeta_5$ reduce to the Cartan coboundary $\zeta_4$ on the simplex $\langle 12345 \rangle$: $\zeta_4(\bar{A}, B)(12345) \equiv \bar{A}(12)\bar{A}(23)B(345)B(235) = \bar{A} \cup [(\bar{A} \cup B) \cup_2 B + \bar{A} \cup B](12345)$, which is used to compute the $O_4$ anomaly in (2+1)D fermionic invertible phases [35].

We use $Sq^2(dn_3) = dn_3 \cup_2 dn_3 \overset{2}{=} d(n_3 \cup_1 n_3 + n_3 \cup_2 dn_3)$ and define $m_2 = [n_2 + \omega_2]_2$. The above identity becomes

$$d(n_3 \cup_1 n_3 + n_3 \cup_1 dn_2) \overset{2}{=} m_2 \cup m_2 \cup n_2 + \frac{dm_2}{2} \cup \frac{dn_2}{2} + m_2 \cup n_2 \cup n_2 + d\zeta_5(m_2, n_2)$$

$$\overset{2}{=} d(m_2 \cup n_3) + d\left(-\frac{1}{2} m_2 \cup (n_2 \cup_1 n_2)\right) + d(n_3 \cup n_2) + d\zeta_5(m_2, n_2) \tag{143}$$

$$\overset{2}{=} d\left(\omega_2 \cup n_3 + dn_3 \cup_1 n_2 - \frac{1}{2} m_2 \cup (n_2 \cup_1 n_2) + \zeta_5(m_2, n_2)\right) .$$

Therefore, we define our $O_5$ as

$$O_5 \equiv \frac{1}{2}\left(n_3 \cup_1 n_3 + n_3 \cup_2 dn_3 + \omega_2 \cup n_3 + dn_3 \cup_1 n_2 + \zeta_5(n_2 + \omega_2, n_2)\right) + \frac{1}{4}[n_2 + \omega_2]_2 \cup (n_2 \cup_1 n_2) . \tag{144}$$

Notice that this is the minimal choice for the $O_5$ obstruction. In general, adding a cocycle to $O_5$ is possible.

## C.1  Comparison with the literature

Let us show that the anomaly $O_5$ obtained agrees with the result in Eq. (220) of Ref. [34].

The Eq. (220) of Ref. [34] expressed the $O_5$ obstruction ($n_1 = 0$) as

$$\mathcal{O}_5(012345) = (-1)^{\omega_2 \cup n_3 + n_3 \cup_1 n_3 + n_3 \cup_2 dn_3(012345) + \omega_2(013)dn_3(12345)}$$

$$\times (-1)^{dn_3(02345)dn_3(01235) + \omega_2(023)[dn_3(01245) + dn_3(01235) + dn_3(01234)]} \tag{145}$$

$$\times i^{dn_3(01245)dn_3(01234) \ (\mathrm{mod}\ 2)}(-i)^{[dn_3(12345) + dn_3(02345) + dn_3(01345)]dn_3(01235)} .$$

We are going to simplify it step by step:

1. $(-1)^{dn_3(02345)dn_3(01235)}$: this term is

$$\frac{1}{2} m_2(023)n_2(345)m_2(012)n_2(235) \overset{1}{=} \frac{1}{2}\zeta_5(m_2, n_2) , \tag{146}$$

where we have used $dn_3(02345) \overset{2}{=} m_2(023)n_2(345)$ and $dn_3(01235) \overset{2}{=} m_2(012)n_2(235)$.

2. $(-1)^{\omega_2(013)dn_3(12345) + \omega_2(023)[dn_3(01245) + dn_3(01235) + dn_3(01234)]}$: this term is

$$\frac{1}{2}\left(\omega_2(013)m_2(123)n_2(345) + \omega_2(023)m_2(012)[n_2(245) + n_2(235) + n_2(234)]\right)$$

$$\overset{1}{=} \frac{1}{2}(\omega_2(013)m_2(123) + \omega_2(023)m_2(012))n_2(345)$$

$$\overset{1}{=} \frac{1}{2}(m_2(013)n_2(123) + m_2(023)n_2(012))n_2(345)$$

$$+ \frac{1}{2}(n_2(013)n_2(123) + n_2(023)n_2(012))n_2(345)$$

$$+ \frac{1}{2}(\omega_2(013)\omega_2(123) + \omega_2(023)\omega_2(012))n_2(345)$$

$$\overset{1}{=} \frac{1}{2}(m_2(013)n_2(123) + m_2(023)n_2(012))n_2(345) + \frac{1}{2}(n_2 \cup_1 n_2) \cup n_2 + \frac{1}{2}(\omega_2 \cup_1 \omega_2) \cup n_2 \tag{147}$$

$$\sim \frac{1}{2}(m_2(013)n_2(123) + m_2(023)n_2(012))n_2(345) + \frac{1}{2}n_2 \cup (n_2 \cup_1 n_2) + \frac{1}{2}\omega_2 \cup (n_2 \cup_1 n_2)$$

$$\overset{1}{=} \frac{1}{2}(m_2(013)n_2(123) + m_2(023)n_2(012))n_2(345) + \frac{1}{2}m_2(012)(n_2(235)n_2(345) + n_2(245)n_2(234))$$

$$\overset{1}{=} \frac{1}{2}(dn_3(01235)n_2(345) + dn_3(01245)n_2(234) + dn_3(01345)n_2(123) + dn_3(02345)n_2(012))$$

$$\overset{1}{=} \frac{1}{2}dn_3 \cup_1 dn_2(012345) .$$

3. $i^{dn_3(01245)dn_3(01234) \ (\mathrm{mod}\ 2)}(-i)^{[dn_3(12345) + dn_3(02345) + dn_3(01345)]dn_3(01235)}$: the first part is

$$\frac{1}{4} m_2(012)n_2(245)n_2(234) . \tag{148}$$

The second part is

$$-\frac{1}{4}[m_2(123) + m_2(023) + m_2(013)]_2 \cdot n_2(235)m_2(012)n_2(235) = -\frac{1}{4}m_2(012)n_2(235)n_2(235) \ . \qquad (149)$$

Combining the above equations, it is simply

$$\frac{1}{4}m_2 \cup (n_2 \cup_1 n_2)(012345) \ . \qquad (150)$$

To sum up, Wang-Gu's $O_5$ can be written as

$$O_5 = \frac{1}{2}\left(n_3 \cup_1 n_3 + n_3 \cup_1 dn_2 + \omega_2 \cup n_3 + dn_3 \cup_1 n_2 + \zeta_5(m_2, n_2)\right) + \frac{1}{4}m_2 \cup (n_2 \cup_1 n_2) \ , \qquad (151)$$

which is exactly Eq. (144) derived in the previous section.

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
