# Peer review of "Higher-group symmetry in finite gauge theory and stabilizer codes"

_SciPost Physics_

## Round 1 · Referee Report · Anonymous · 2023-9-22

Strengths
-detailed study of higher-group symmetry in finite gauge theories and several exciting applications, such as the classification of fermionic SPTs and fault-tolerant gates
-very well written and clear
Weaknesses
-outlook with possible future directions could be mentioned
Report
This paper comprehensively studies the higher group symmetry and its 't Hooft anomaly in finite gauge theories. It describes the higher-group symmetry as arising from a generalization of the Witten effect and the charge-flux attachment. It provides several explicit examples, including field theories and lattice models. The applications discussed in the paper are exciting. For instance, studying the fault-tolerant logical gates from this perspective could lead to insights for practical applications.
The paper's presentation and results are excellent. I wholeheartedly recommend publishing it in SciPost Physics.

---

## Round 1 · Referee Report · Anonymous · 2024-1-9

Strengths
1. Clearly written and systematic.
2. Many concrete examples are presented.
Report
The paper studies the invertible symmetries of Dijkgraaf-Witten gauge theories in all dimensions. It is clearly written with numerous examples presented to illustrate the conclusions. I recommend the paper wholeheartedly for publication with minimal changes. My suggestions pertain only to setting the results in a mathematical context.
- My understanding is that Dijgraaf-Witten theory for a finite group $G$ in D = d+1 dimensions should have $(D-1)$-fusion category symmetry (in the bosonic case and putting aside issues of unitarity) $\Sigma Z((D-2)Vect_G^{\omega_D})$ where $Z$ denote the Drinfeld center and \Sigma condensation completion of the de-looping. Then from a mathematical perspective, I expect the authors are computing the invertible part of this symmetry category. Is that correct?
- Following from the above, I expect general genuine codimension-2 defects (genuine = not attached to a codimension-1 topological defect) are labelled by a conjugacy class in $G$ and a projective $(D-2)$-representation of the centraliser with projective $(D-1)$-cocyle given by the transgression of $\omega_D$. This seems compatible with the discussion at the beginning of section 2, and includes magnetic defects stacked with gauged SPT phases for the unbroken subgroup. However, it may be nice to relate the discussion in 2,2.1 to this mathematical classification. (I appreciate starting with 2.2 the author's consider instead twisted sector magnetic defects attached to codimension-1 topological defects.)
Requested changes
I suggest some discussion of how the results relate to the aforementioned mathematical framework. However, as this is primarily a physics oriented journal, this should be taken as a suggestion only!

---

## Editorial Decision

unknown